# Osteology of the Hamadryas Baboon (*Papio hamadryas*)

**DOI:** 10.3390/ani13193124

**Published:** 2023-10-06

**Authors:** Christophe Casteleyn, Estée Wydooghe, Jaco Bakker

**Affiliations:** 1Department of Morphology, Medical Imaging, Orthopedics, Physiotherapy and Nutrition, Faculty of Veterinary Medicine, Ghent University, Salisburylaan 133, 9820 Merelbeke, Belgium; estee.wydooghe@gmail.com; 2Department of Veterinary Sciences, Faculty of Pharmaceutical, Biomedical and Veterinary Sciences, University of Antwerp, Universiteitsplein 1, 2610 Wilrijk, Belgium; 3Animal Science Department, Biomedical Primate Research Centre, Lange Kleiweg, 161, 2288GJ Rijswijk, The Netherlands; bakker@bprc.nl

**Keywords:** anatomy, baboon, bones, non-human primate, osteology, skeleton

## Abstract

**Simple Summary:**

The skeleton of a mammal plays an important role in the support of the body. It allows movement, as ligaments and muscles are attached to specific bony structures. In addition, nerves and blood vessels pass through osseous openings and canals. Finally, organs are protected by the skeleton. These skeletal functions are exemplified by the skull, which protects the brain. Nerves and blood vessels run from and towards the brain through numerous openings. The skull provides a joint with the mandible, allowing to chew food by means of the masticatory muscles. Portraying the anatomy of a mammal therefore starts with the description of its skeleton. Here, we describe the hamadryas baboon (*Papio hamadryas*), a popular primate in zoos and can be found in research centers as well. Although its anatomy has already been described, the anatomical works are outdated, incomplete, use archaic nomenclature and fail to provide high-definition color photographs. We have revisited the skeletal anatomy of the baboon by photographing its bones and labelling these using the latest edition of the veterinary anatomical terminology list. We provide 31 annotated multipanel figures that can serve as a basis for further anatomical research on the hamadryas baboon.

**Abstract:**

Besides living as a free-ranging primate in the horn of Africa and the Arabian Peninsula, the hamadryas baboon has an important place in zoos and can be found in biomedical research centers worldwide. To be valuable as a non-human primate laboratory model for man, its anatomy should be portrayed in detail, allowing for the correct interpretation and translation of obtained research results. Reviewing the literature on the use of the baboon in biomedical research revealed that very limited anatomical works on this species are available. Anatomical atlases are incomplete, use archaic nomenclature and fail to provide high-definition color photographs. Therefore, the skeletons of two male hamadryas baboons were prepared by manually removing as much soft tissues as possible followed by maceration in warm water to which enzyme-containing washing powder was added. The bones were bleached with hydrogen peroxide and degreased by means of methylene chloride. Photographs of the various bones were taken, and the anatomical structures were identified using the latest version of the Nomina Anatomica Veterinaria. As such, the present article shows 31 annotated multipanel figures. The skeleton of the hamadryas baboon generally parallels the human skeleton, but some remarkable differences have been noticed. If these are taken into consideration when evaluating the results of experiments using the hamadryas baboon, justified conclusions can be drawn.

## 1. Introduction

The study of human pathology, including the unraveling of disease mechanisms, and the development of curative and prophylactic strategies to treat and prevent disease is facilitated by means of animal models [1]. For the obtained results to be valuable and relevant, the animal model of choice should be closely related to man [2]. Due to the very high genetic similarities between non-human primates and humans, and given the fact that research on human primates such as chimpanzees is fraught by huge ethical concerns, non-human primate species, such as marmosets, rhesus monkeys and baboons, can be found in numerous research facilities worldwide [3,4,5,6].

The relationship between humans and baboons can be traced back to ancient Egypt. Although they are not indigenous to Egypt, baboons certainly lived in that region. This has been concluded after baboon skeletons had been discovered in ancient, sacred sites in Egypt. These animals were thus traded as luxury goods around the Red Sea. They were, however, kept in inappropriate conditions since rickets and arthrosis have frequently been encountered in the discovered baboon skeletons [7]. Evidence exists that captive breeding programs had been installed in an attempt to be self-sufficient. This was unfortunately not very successful [8]. The Egyptians apparently lacked the knowledge to provide the correct care for the animals. In addition, the imported baboons also posed a threat to public health as they introduced zoonotic parasitic diseases [9]. On the other hand, the baboon was one of the animal species into which gods might have been transformed in ancient Egypt. In particular, *Papio hamadryas*, the baboon species that is examined in the present manuscript, was worshipped, hence its alternative name, ‘sacred baboon’. They were indeed sacred. After their death at a maximal age of around six years, the cadavers were mummified. This is known since mummies have been recovered from Egyptian temples and tombs [8].

For decades, the baboon was a preferred non-human primate research subject. As an animal model, it served in research on, amongst others, reproduction, chronic pulmonary diseases, osteoporosis, cardiovascular diseases including hypertension and atherosclerosis, and the development of vaccines against hepatitis C and HIV [10,11,12]. Its popularity started to decline at the beginning of the new millennium, since the large body size requires considerable volumes of food and adequately built cages with outdoor runs, which is expensive [10]. However, a revival in the use of the baboon for research purposes has been observed more recently [13]. This species appears to be a valuable model to investigate the metabolic syndrome e.g., [14], liver diseases e.g., [15] or cardiovascular diseases e.g., [16]. In addition, the baboon has proven to be very instrumental in the study of infectious, zoonotic diseases evoked by microorganisms such as *Balantidium coli* [17], genital papillomavirus [18], middle east respiratory syndrome coronavirus (MERS-CoV) [19] and, not to forget, SARS-CoV-2 [20].

The genus *Papio* includes six species: *Papio anubis*, *Papio cynocephalus*, *Papio hamadryas*, *Papio kindae, Papio papio*, and *Papio ursinus*, which all belong to African wildlife [10,21]. *Papio hamadryas*, the hamadryas, mantled or sacred baboon is a non-endangered Old World monkey that finds its natural habitats in the regions around the Red Sea [21]. In contrast to the majority of non-human primates that are arboreal, the hamadryas baboon is a terrestrial ranger [22]. It is omnivorous, feeding on grass seeds, Acacia legumes, leaves, and roots. The diet is supplemented with insects, and sporadically a hare or infant antelope is on the menu [22,23]. The sacred baboon presents sexual dimorphism with the male being bigger than the female. The male’s body measures approximately 65 cm but can reach a length up to 80 cm, accounting for a weight of 20 to 30 kg, whereas females present barely two thirds to half the male’s weight (10 to 15 kg) and are only half the size. The baboon’s body is elongated by the tail that measures 40 to 60 cm in length, presenting a tufted tip [24]. In addition to the discriminating body size, males are characterized by a grey coat with a typical cape (mane and mantle) that consists of longer, silver hairs. The females lack such a cape and are uniformly brown. They fail to exhibit external distinguishing traits. In either sex, the face and ischial region are hairless and colored red to brown [22,25]. The latter region is characterized by a pair of pronounced ischial callosities in both sexes. During the estrus in females, the ischial region is characteristically red and swollen [26,27]. Hamadryas baboons can reach an age of 25 to 30 years, both in the wild as in captivity [28].

It may be deduced from the paragraph above that abundant data can be found in the literature on the genetics, ecology, behavior, etc., of the hamadryas baboon. However, if baboons ought to be used in studies that demand a comparable anatomy with man, a solid knowledge of its anatomy is prerequisite for the correct interpretation of experimental results. In addition, comprehension of the baboon’s anatomy can also be pivotal for veterinarians charged with the daily medical care and welfare of captive baboons. Unfortunately, literature on the anatomy of the baboon is rather limited. Most anatomical works that are available focus on a specific topic—e.g., the brain and female reproductive tract are intensively studied—and fail to provide a general overview of the baboon’s anatomy. Only *An Atlas of Primate Gross Anatomy: Baboon, Chimpanzee and Man* [29] reaches that goal to a high extent. As the title suggests, this atlas is primarily designed to enhance the anatomical comparison between the baboon, the chimpanzee and man. As a result, this atlas has a few shortcomings as regards the in-depth anatomical description of each single species. The osteology is described rather superficially with only the major anatomical structures labelled. The study of the joints (arthrology) is not covered. The myology (study of the musculature) is not fully elaborated but the figures that are presented are profound. The nomenclature used is sometimes archaic and needs to be updated to the standards of the Nomina Anatomica Veterinaria [30]. In addition, some typos and erroneously labelled structures can be noticed. Finally, the figures are black and white line drawings that do not fully represent the complexity of the anatomy.

The aim of the present study was to revisit the anatomy of the baboon as described by Swindler and Wood [29]. Since it is impossible to portray the entire anatomy in a single manuscript, the focus of this work is the osteology, which is the study of the skeleton. The atlas [29] served as a starting point for the present manuscript. During the description of the baboon skeleton, reference is made to that atlas in order to clarify the function of some prominent osteological structures. A few other works have been consulted during the preparation of this study. The manuscript by Fleagle and McGraw [31] was valuable for its comparison of the scapula, humerus, radius, ulna, pelvis, femur, and teeth between several species of the genus *Papio*. Other works were contributory as they depict the anatomy of the vertebral column [32]. Cranial anatomy of the baboon is illustrated by Trevor-Jones [33] and [34]. Furthermore, we made use of our previous works on the anatomy of the marmoset (*Callithrix jacchus*) and rhesus monkey (*Macaca mulatta*) to identify the numerous anatomical structures in the baboon by means of comparative anatomy between primate species [35,36,37,38,39].

## 2. Materials and Methods

### 2.1. Animals

The cadavers of three adult hamadryas baboons, two males and one female, were used in this study. They were obtained from the Biobank of the Biomedical Primate Research Centre (BPRC), Rijswijk, The Netherlands (https://www.bprc.nl/en/biobank, last accessed 8 September 2023). After transportation to the Faculty of Veterinary Medicine, Ghent University, Belgium, the cadavers were stored at −20 °C. Prior to the anatomical examinations, the cadavers were thawed at room temperature.

### 2.2. Preparation Techniques and Imaging

First, the thawed cadavers were deskinned, and their musculature was studied in the framework of another study. After the dissection of the muscles, the cadavers were manually defleshed. Subsequently, the soft tissue remnants were macerated in 20% sodium hypochlorite (NaOCl^-^) (Brenntag n.v., Deerlijk, Belgium). Maceration was, however, terminated once the ligamentous structures associated with the joints and vertebral column commenced to deteriorate. This maceration technique allowed for the arthrological examination that is part of another study. Next, the specimens were each immersed in 50 L warm water (60 °C) to which 200 g Biotex^®^ washing powder (Unilever Nederland B.V., Rotterdam, The Netherlands) was added for one week. This procedure digested all soft tissues resulting in the disconnection of the skeletal structures. These were then rinsed with tap water, and for one week were immersed in 10% hydrogen peroxide (H_2_O_2_) (Brenntag n.v., Deerlijk, Belgium) to bleach the bones. Next, the bones were rinsed again. Finally, the bones were degreased in a distillation process using methylene chloride (Brenntag n.v., Deerlijk, Belgium). As such, all bony structures could be examined in the osteological study.

The bones of the skeleton were illustrated by means of photographs taken with a digital photo camera (Canon EOS1300D, Canon Belgium, Diegem, Belgium). To this purpose, the specimens were positioned onto a black background. Afterwards, the pictures were manipulated in GIMP 2.10.30 (www.gimp.org, accessed on 3 January 2023). The background was rendered equally black, contrast and brightness were adjusted where appropriate, and the anatomical structures were labelled.

The length and width of the skull, the length of the mandible, the length of each vertebral segment, the length of the sternum, the length of the left clavicle, the length and width of the bony pelvis, the lengths of the long bones of the left appendicular skeleton, and the lengths of the skeletons of the left hand and foot were determined by means of measuring tape and digital calipers.

### 2.3. Anatomical Nomenclature

The anatomical terms that are applied in this article are derived from the Nomina Anatomica Veterinaria [30]. This reference work is, however, intended to be used when describing the anatomy of domestic mammals and does therefore not provide specific terms for the baboon. The anatomical studies on the baboon that have been cited above in the introduction to our work use human terminology. As a non-human primate, the baboon is not anatomically identical to humans. Thus, not all human terms are applicable to the baboon. Employing human anatomical nomenclature to describe the anatomy of a non-human species can be confusing and hampers the comparison between different anatomical works. In our work, we opted to use veterinary nomenclature as much as possible. When a veterinary anatomical term is not available to describe a structure that is typical for primates, the human term was applied.

When textually describing the results of our anatomical study, the English terms of the various anatomical structures are followed by the Latin terms between brackets the first time a structure is mentioned. Solely, English terminology is then applied in the further elaboration of the structure to increase the readability of the text. In contrast, only Latin terminology is employed in the legends associated with each figure. For easy use of the figure legends, the Latin anatomical terms are presented in alphabetical order.

## 3. Results

### 3.1. Skeleton in General

The mounted skeleton of one of the two male examined hamadryas baboons is presented in Figure 1. In the text below, the various components of the skeleton are described in detail. Here, we give a general overview of the skeleton (*systema skeletale*), which includes the axial skeleton (*skeleton axiale*) and the appendicular skeleton (*skeleton appendiculare*).

The axial skeleton comprises the skeleton of the head, the vertebral column (*columna vertebralis*), and the bony thorax (*skeleton thoracis*). The skeleton of the head is composed of the skull (*cranium*) and the mandible (*mandibula*). The vertebral column consists of the cervical segment that contained seven cervical vertebrae (*vertebrae cervicales*) in the three examined specimens, the thoracic segment comprised twelve thoracic vertebrae (*vertebrae thoracicae*) in the three examined specimens, the lumbar segment held seven lumbar vertebrae (*vertebrae lumbales*) in the three examined specimens, the sacrum (*os sacrum*) was composed of three fused sacral vertebrae (*vertebrae sacrales*) in the three examined specimens and, finally, the tail had twenty caudal vertebrae (*vertebrae caudales* or *vertebrae coccygeae*) in the two male skeletons and twenty-one in the female skeleton. The bony thorax includes the thoracic vertebrae, the ribs (*costae*) and the sternum (*sternum*), and as such forms the thoracic cage (*cavum thoracis*). The latter has a cranial entrance (*apertura thoracis cranialis*) formed by the first thoracic vertebra dorsally, the first pair of ribs laterally and the manubrium of the sternum (*manubrium sterni*) ventrally. The caudal exit of the thoracic cage (*apertura thoracis caudalis*) is much larger as it is formed by the last thoracic vertebra, the last pair of ribs and the xyphoid process (*processus xiphoideus*) of the sternum.

The appendicular skeleton contains the bones of the thoracic limb (*ossa membri thoracici*) and the bones of the pelvic limb (*ossa membri pelvini*). The thoracic limb is attached to the thorax by means of the shoulder or pectoral girdle (*cingulum membri thoracici*), which is composed of the shoulder blade (*scapula*) and the collar bone or clavicle (*clavicula*). The brachium (*skeleton brachii*) consists of the humerus (*humerus*). This bone forms the stylopodium of the thoracic limb. The antebrachium (*skeleton antebrachii*) holds the radius (*radius*) and the ulna (*ulna*). These bones form the zeugopodium of the thoracic limb. The skeleton of the hand (*skeleton manus*) contains the carpal bones (*ossa carpi*), the metacarpal bones I-V (*ossa metacarpalia I-V*) and the phalanges (*ossa digitorum manus*) with their associated sesamoid bones (*ossa sesamoidea palmaria*). The pelvic limb is attached to the vertebral column by means of the pelvic girdle (*cingulum membri pelvini*). This pelvic girdle is formed by the hip bones (*ossa coxae*). Each (left and right) hip bone (*os coxae*) is composed of the fused ilium (*os ilium*), ischium (*os ischii*) and pubis (*os pubis*). The left and right hip bones are fused at the pelvic symphysis (*symphysis pelvina*), and together with the sacrum form the hip (bony pelvis). The femoral skeleton (*skeleton femoris*) contains the femur (*os femoris* or simply *femur*), the kneecap (*patella*) and the sesamoid bones associated with the knee (*ossa sesamoidea musculi gastrocnemii*). The femur forms the stylopodium of the pelvic limb. The crural skeleton (*skeleton cruris*) comprises the tibia (*tibia*) and the fibula (*fibula*). These bones constitute the zeugopodium of the pelvic limb. The foot skeleton (*skeleton pedis*) contains the tarsal bones (*ossa tarsi*), the metatarsal bones I-V (*ossa metatarsalia I-V*) and the phalanges (*ossa digitorum pedis*) with their associated sesamoid bones (*ossa sesamoidea plantaria*).

A diminutive, slightly curved penile bone (*os penis*) is present in the male hamadryas baboon (Figure 1B, insert). It measured 1.7 cm in length and 2.2 mm in width in male skeleton 1, and 1.8 cm in length and 2.3 mm in width in male skeleton 2.

### 3.2. Skull (Cranium)

#### 3.2.1. General Conformation

Figure 2 presents a general overview of the skull. It measured 18.7 cm in length and 11.8 cm in width in male skeleton 1, and 19.2 cm in length and 12.1 cm in width in male skeleton 2. The values for the female were 16.7 cm and 9.9 cm, respectively.

The facial bones of the skull (*ossa faciei*) form the *splanchnocranium*, while the cranial bones of the skull (*ossa cranii*) constitute the *neurocranium*. The *splanchnocranium* is longer than the cranial skeleton and lies somewhat more ventral than the *neurocranium*, a trait that has maximally evolved in man [29]. The different bones of the facial and cranial skeleton are joined by sutures and synchondroses, which are particularly visible on the neonatal skull. A suture (*sutura*) is a fibrous articulation (*articulatio fibrosa*), whereas a synchondrosis is a cartilaginous articulation (*articulatio cartilaginea*). Not all sutures were easily identified in the adult baboon. Most sutures were no longer cartilaginous in nature but could be identified as bony ridges.

The facial bones of the baboon, which form the *splanchnocranium*, include the paired nasal bone (*os nasale*), the paired maxillary bone (*maxilla*), the paired lacrimal bone (*os lacrimale*), the paired ventral nasal conchal bone (*os conchae nasalis ventralis*), the paired incisive bone (*os incisivum*), the paired palatine bone (*os palatinum*), and the paired zygomatic bone (*os zygomaticum*). The cranial bones that form the braincase include the paired occipital bone (*os occipitale*), the paired basisphenoid bone (*os basisphenoidale*), the paired presphenoid bone (*os presphenoidale*), the paired pterygoid bone (*os pterygoideum*), the paired temporal bone (*os temporale*), the paired parietal bone (*os parietale*), the frontal bone (*os frontale*), which is initially paired but forms a single bone after the frontal or metopic suture (*sutura frontalis*) has disappeared at juvenile age [29], the paired ethmoidal bone (*os ethmoidale*), and the single vomer.

On the lateral view of the skull presented in Figure 2, the highlighted border between the *splanchnocranium* and the *neurocranium* is formed by the frontonasal suture (*sutura frontonasalis*), the frontomaxillary suture (*sutura frontomaxillaris*), the frontolacrimal suture (*sutura frontolacrimalis*), the ethmoidolacrimal suture (*sutura ethmoidolacrimalis*), the ethmoidomaxillary suture (*sutura ethmoidomaxillaris*), the sphenozygomatic suture (*sutura sphenozygomatica*), the frontozygomatic suture (*sutura frontozygomatica*), and the temporozygomatic suture (*sutura temporozygomatica*).

#### 3.2.2. Orbit (*Orbita*)

A very prominent structure at the transition of the *splanchnocranium* to the *neurocranium* is the orbit (*orbita*) (Figure 3). It is composed of multiple facial and cranial bones. The roof of the orbit (*paries dorsalis*) is formed by the orbital part of the frontal bone (*pars orbitalis ossis frontalis*) and the wing of the presphenoid bone (*ala ossis presphenoidalis*). The wing of the basisphenoid bone (*ala ossis basisphenoidalis*) and the zygomatic bone constitute the lateral orbital wall (*paries lateralis*). The medial orbital wall (*paries medialis*) is composed of the wing of the basisphenoid bone (*ala ossis basisphenoidalis*), the orbital plate of the ethmoidal bone (*lama orbitalis ossis ethmoidalis*), the lacrimal bone and the frontal process of the maxilla (*processus frontalis maxillae*). The floor of the orbit (*paries ventralis*) constitutes the orbital process of the palatine bone (*processus orbitalis ossis palatini*), the maxilla and the zygomatic bone. 

The *nervus* (*n.*) *opticus* reaches the cranial cavity (*cavum cranii*) through the optic foramen (*foramen opticum*), which is the dorsomedial opening of the short but wide dorsal orbital fissure (*fissura orbitalis dorsalis*) (superior orbital fissure, *fissura orbitalis superior* in man) [29]. It leads to the optic canal (*canalis opticus*) within the wing of the presphenoid bone. The *n. oculomotorius*, the *n. trochlearis* and the *n. abducens* reach the orbit via the caudal extremity of the dorsal orbital fissure, lateral to the optic foramen [29]. We denote this opening as the orbital foramen (*foramen orbitale*). The three nerves are joined by the *n. ophthalmicus* of the *n. trigeminus* [29]. The *n. maxillaris* leaves the cranial cavity via the round foramen (*foramen rotundum*) that can be observed in the depth, caudoventral in the longer and ventrally positioned orbital fissure (*fissura orbitalis ventralis*) (inferior orbital fissure, *fissura orbitalis inferior* in man) [29]. This fissure, which is caudally continuous with the dorsal orbital fissure, divides the bottom of the orbit in a medial and a lateral compartment. Medial to this fissure is the opening to the nasal cavity. Laterally, there is communication with the pterygopalatine fissure (*fissura pterygopalatina*). The rostral extremity of the ventral orbital fissure prolongs since the infraorbital groove (*sulcus infraorbitalis*) leads to multiple infraorbital foramens (*foramina infraorbitalia*, singular: *foramen infraorbitale*), just ventral tot the ventral margin of the orbit. Here, the *n. infraorbitalis* (from the *n. maxillaris*) and *arteria (a.) infraorbitalis* emerge [29]. In the orbit, a branch of the maxillary nerve (*ramus zygomaticus*) divides into the *n. zygomaticotemporalis* that runs close to the lateral orbital wall towards the zygomaticotemporal foramen (*foramen zygomaticotemporale*), and the *n. zygomaticofacialis*, which travels ventrolaterally in the orbit towards the zygomaticofacial foramen (*foramen zygomaticofaciale*) [29].

In the hamadryas baboon, the lacrimal bone lies completely within the orbital margin. The *fossa sacci lacrimalis* is a depression in the lacrimal bone, situated at the medial side of the orbit, which leads to the nasolacrimal duct (*ductus nasolacrimalis*). An accessory duct can be seen ventrolateral to the entrance of this duct. The rostral border of the lacrimal sac is sharp and known as *crista lacrimalis rostralis*.

#### 3.2.3. *Splanchnocranium*

In contrast to the ventral margin of the orbit (*margo infraorbitalis*), the dorsal margin (*margo supraorbitalis*) is pronounced. The supraorbital ridge (*arcus supraciliaris*) presents a supraorbital torus (*torus supraorbitalis*) in its central aspect, lateral to the supraorbital notch (*incisura supraorbitalis*) (Figure 3 and Figure 4A). The *glabella* in between both supraorbital ridges protrudes rostrally (Figure 4B). Here, remnants of the interfrontal or metopic suture (*sutura interfrontalis*) can be noticed (Figure 4D).

More caudally, at the dorsal aspect of the skull, this suture was no longer visible in the examined adult baboons (Figure 4B). However, the major cranial sutures, i.e., the sagittal suture (*sutura sagittalis*) between the parietal bones, the coronal suture (*sutura coronalis*) between the frontal and parietal bones, and the lambdoid suture (*sutura lambdoidea*) between the parietal and occipital bones were easily recognized (Figure 4B,E). The *bregma* can be identified as the junction of the coronal suture and the sagittal suture. It is located just rostral to the *vertex*, which is the highest point of the *calvaria*, the roof of the skull (Figure 4B). The area where the parietal, temporal, frontal and sphenoid bones adjoin is called *pterion* (Figure 4A).

The *nasion* is the most rostral point of the frontonasal suture. It is vertically oriented (Figure 4A). Caudally, the bilateral nasal bones that are joined by the internasal suture (*sutura internasalis*) are very slender and form a sharp nasal bridge (*dorsum nasi*). The internasal suture itself was no longer visible in the adult baboons that were examined, but a pronounced bony ridge was obvious at the ventral side of this suture (Figure 4D). The nasal bones widen and flatten more rostrally (Figure 4B,D). The nasal process (*processus nasalis*) of the incisive bone forms the nasoincisive suture (*sutura nasoincisiva*) where it adjoins the broader, rostral aspects of the ipsilateral nasal bone (Figure 2). A small indentation, the nasoincisive notch (*incisura nasoincisiva*), is present at the rostral extremity of this suture (Figure 4A). The nasal processes of the incisive bones, together with the nasal bones, form the osseous nasal opening (*apertura nasi ossea*) (Figure 4D).

The suture between the maxilla and the incisive bone (*os incisivum* or *premaxilla* in man) (*sutura maxilloincisiva* or *sutura premaxillomaxillaris* in man) was obvious [29]. Rostrally, it was located in the *diastema* between the second incisor (*dens incisivus secundus*) and the caninus (*dens caninus*), close to the mesial side (*facies mesialis*) of the caninus (Figure 2 and Figure 4C,D). The incisive bone presents two incisors, i.e., the first incisor (*dens incisivus* (*maxillaris*) *primus*) and the second incisor (*dens incisivus* (*maxillaris*) *secundus*) (Figure 4C,D). The maxilla is large and bears the remainder of the maxillary set of teeth. To this purpose, it presents the alveolar process (*processus alveolaris*), which contains the dental alveoli (singular: *alveolus*). The canine is prominent, in particular in the male. Then, two premolars, i.e., the third premolar (*dens premolaris* (*maxillaris*) *tertius*) and the fourth premolar (*dens premolaris* (*maxillaris*) *quartus*) precede three molars, i.e., the first molar (*dens molaris* (*maxillaris*) *primus*), the second molar (*dens molaris* (*maxillaris*) *secundus*), and the third molar (*dens molaris* (*maxillaris*) *tertius*). The maxilla shows an excavation dorsal to the molar alveoli, called the maxillary fossa (*fossa maxillaris*). The crescent arrangement of teeth in each upper jaw is the superior dental arch (*arcus dentalis superior*). Each superior dental arch contains eight teeth (Figure 4C).

The osseous roof of the oral cavity or, in other words, the osseous core of the hard palate (*palatum durum*), which is termed the osseous palate (*palatum osseum*), is formed by the incisive bone, the palatine process (*processus palatinus*) of the maxilla and the horizontal plate of the palatine bone (*lamina horizontalis ossis palatini*). A large opening is present rostrally in the hard palate, in between the incisive bone and the maxilla. This is the incisive foramen (*foramen incisivum*), which allows for the passage of several structures such as the *nervi (nn.) pterygopalatini*, the *nn. nasopalatini*, and the *a.* and *v. sphenopalatina* [29]. The caudal junction or suture between the bilateral incisive bones, i.e., the interincisive suture (*sutura interincisiva*), projects the caudally oriented rostral nasal process (*processus nasalis rostralis*) into the incisive foramen. At the level of the third maxillary molar, the transverse palatine suture (*sutura palatina transversa*) is situated between the maxillary palatine process and the horizontal plate of the palatine bone. From the midline, where the longitudinal median palatine suture joins the bilateral maxillary palatine processes and the horizontal plates of the palatine bones, the transverse palatine suture travels caudolaterally towards the caudal aspect of the maxillary tuberosity (*tuber maxillae*). Medial to this tuberosity, immediately caudal to the transverse palatine suture, is the rostral opening of the greater palatine canal (*canalis palatinus major*) (or pterygopalatine canal), which is denoted as the greater palatine foramen (*foramen palatinum majus*). The *a. palatina descendens* and the *n. palatinus major* emerge here and follow the palatine groove (*sulcus palatinus*) in a rostral direction at the lingual side (*facies lingualis*) of the maxillary teeth where the alveolar process (*processus alveolaris*) of the maxilla borders the palatine process [29]. The above-described anatomical structures can be studied using Figure 4C.

Further elaborating Figure 4C allows for studying the choanal region. Caudal to the horizontal plate of the palatine bone are the *choanae* or caudal nasal apertures, which allow for the communication between the nasal cavity (*cavum nasi*) and the *nasopharynx*. They are separated from each other in the median plane by the caudal nasal process (*processus nasalis caudalis*) of the horizontal plate of the palatine bone, and the *vomer*. This small, unpaired bone, resembling a ploughshare, which is located in the midline of the skull, forms the osseous caudal part of the nasal septum (*septum nasi*). It sits on the body of the sphenoid bone. The lateral border of each choana is the vertical plate of the palatine bone (*lamina verticalis ossis palatini*), and the medial plate of the pterygoid process (*lamina medialis processus pterygoidei*) of the sphenoid bone. This lateral border is reinforced by the pterygoid bone, which is a small osseous lamella adjoined to the medial plate of the sphenoid bone by means of the pterygosphenoidal suture (*sutura pterygosphenoidalis*) and the vertical plate of the palatine bone by means of the pterygopalatine suture (*sutura pterygopalatina*) [33]. At the rostroventral portion of the medial plate is a protuberant hamular process (*hamulus pterygoideus*). Immediately rostral to the hamulus sits the pyramidal process (*processus pyramidalis*) of the palatine bone. Medial to the medial plate sits the deep pterygoid fossa (*fossa pterygoidea*), which is bordered laterally by the lateral plate of the pterygoid process (*lamina lateralis processus pterygoidei*). This lateral plate, which fans out laterally, is massive compared to the minute medial plate.

#### 3.2.4. *Neurocranium*

In Figure 4C, which shows a ventral view of the skull, the sutures forming the borderline between the *splanchnocranium* and the *neurocranium*, are highlighted in black, whereas the other sutures are highlighted in grey. The visible sutures between the *splanchnocranium* and the *neurocranium* include the sphenozygomatic suture (*sutura sphenozygomatica*), the sphenopalatine suture (*sutura sphenopalatina*), and the vomeropalatine suture (*sutura vomeropalatina*).

Dorsolateral to the lateral plate of the pterygoid process of the sphenoid bone is the irregular infratemporal fossa (*fossa infratemporalis*) (Figure 4A). It is medial to the zygomatic arch (*arcus zygomaticus*) that is formed by the temporal process (*processus temporalis*) of the zygomatic bone and the zygomatic process (*processus zygomaticus*) of the temporal bone. Both processes meet halfway at the zygomatic arch at the temporozygomatic suture (*sutura temporozygomatica*) (Figure 2). The ventral aspect of the base of the zygomatic process presents the shallow mandibular fossa (*fossa mandibularis*). Caudal displacement of the mandible is prevented by the vertically oriented retroarticular process (*processus retroarticularis*) (Figure 4C). Medial to the mandibular fossa, caudal to the pterygoid fossa, and rostral to the tympanic bulla (*bulla tympanica*), which is the visibly enlarged bony cavity that encloses the middle ear, are the diminutive, laterally positioned spinous foramen (*foramen spinosum*) that transmits blood vessels, and the large, medially located oval foramen (*foramen ovale*) (Figure 4C) [29]. They lie at the caudal extremity of the lateral plate of the pterygoid process of the sphenoid bone and rostral to the tympanic bulla. The *n. mandibularis* reaches the infratemporal fossa via the *foramen ovale* [29]. This foramen thus connects the middle cranial fossa (*fossa cranii media*) with the infratemporal fossa. It is in rostral continuity with the alar canal (*canalis alaris*). This canal has a caudal and a rostral opening, i.e., the caudal alar foramen (*foramen alare caudale*) and the rostral alar foramen (*foramen alare rostrale*), respectively. Since the alar canal is very short, the perforation of the caudolateral surface of the lateral pterygoid plate by the rostral branch of the mandibular nerve is briefly called the pterygoalar foramen (*foramen pterygoalare*), which is rarely present in humans [29]. This foramen is located ventral to the infratemporal crest (*crista infratemporalis*) (Figure 5). The large caudal branch of the mandibular nerve also exits via the oval foramen but travels along the medial side of the pterygoid plate, thus not entering the pterygoalar foramen [29].

The pterygopalatine fossa (*fossa pterygopalatina*) lies deep in the infratemporal fossa (Figure 5). Dorsally located in the pterygopalatine fossa is the sphenopalatine foramen (*foramen sphenopalatinum*) through which the nasal cavity can be reached. The very narrow, slit-like pterygopalatine fissure (*fissura pterygopalatina*) can be observed ventral to this foramen. It lies dorsal to the pyramidal process (*processus pyramidalis*) of the palatine bone that sits in between the maxillary tuberosity and the lateral plate of the pterygoid process. The maxillary artery leaves the masticatory space by way of the pterygopalatine fossa after a number of branches were given off [29]. The pterion region is dorsal to the infratemporal fossa (Figure 2). In man, *pterion* is the area where the parietal, temporal, frontal and sphenoid bones adjoin [29]. In *Papio hamadryas*, the temporal bone articulates with the frontal bone by means of the temporofrontal suture (*sutura temporofrontalis*) (Figure 2 and Figure 4A).

The squamous part of the temporal bone, just caudal to the sphenosquamous suture, is slightly shallow. This is the temporal fossa (*fossa temporalis*). It can thus be found dorsomedial to the zygomatic arch (Figure 4A,B). Dorsal to the supraciliary arch, starting laterally at the zygomaticotemporal foramen, lies the temporal line (*linea temporalis*), where the temporal muscle originates [29]. After bending medially, it runs in caudal direction, parallel to the midline of the skull where the sagittal suture is located (Figure 4B,E). This suture forms the junction between the left and right parietal bones. Here, no external sagittal crest (*crista sagittalis externa*) can be observed in the hamadryas baboon. In contrast, her most caudal extremity presents an osseous thickening known as the external occipital protuberance (*protuberantia occipitalis externa*), which serves as insertion site for the trapezius muscle and the nuchal ligament (*ligamentum nuchae*) [29]. The sharp, highest point is termed the *inion*. This protuberance is located at the junction of the left and right parietal bones, and the left and right occipital squames (singular: *squama occipitalis*) of the occipital bones. These squames make contact in the midline at the interoccipital suture (*sutura interoccipitalis*), which presents a bony ridge termed the external occipital crest (*crista occipitalis externa*) (Figure 4E). She is pronounced at the level of the nuchal plane (*planum nuchale*). This flat region of the skull is separated from the more dorsal, concave region of the occiput by the bilateral inferior nuchal lines (singular: *linea nuchae inferior*) (Figure 4C). Dorsal to this line is the superior nuchal line (*linea nuchae superior*) or the nuchal crest (*crista nuchae*). It is the crest on the lambdoid suture, which is the suture where the parietal and occipital bones form an acute junction. *Lambda* is where the sagittal suture meets the lambdoid suture. The nuchal crest can be followed in a rostral direction up to the external acoustic pore (*porus acusticus externus*). Here, the temporal line joins the termination of the nuchal crest that becomes confluent with the temporal crest (crista temporalis). This is the dorsal, sharp margin of the zygomatic arch (Figure 4A,E).

When studying the *occiput*, the large foramen (*foramen magnum*) cannot be missed. It is the largest opening of the skull, connecting the brain stem with the spinal cord. It is oriented parallel to the longitudinal axis of the skull. The occipital condyles (*condyli occipitales*, singular*: condylus occipitalis*), bilateral to the large foramen, articulate with the atlas. At the basis of each occipital condyle is the foramen through which the hypoglossal nerve exits the cranial cavity (*foramen nervi hypoglossi*) [29]. Caudodorsal to the base of each occipital condyle is a deep indentation, the condylar fossa (*fossa condylaris*). All the above-mentioned structures belong to the basal part (*pars basilaris*) of the occipital bone (Figure 2 and Figure 4C). This basal part slightly narrows in rostral direction to meet the basisphenoid at the sphenooccipital synchondrosis (*synchondrosis sphenooccipitalis*). Immediately lateral to the *foramen nervi hypoglossi* is the jugular fossa (*fossa jugularis*), where the internal jugular vein passes, in between the basal part of the occipital bone and the tympanic part of the temporal bone (*pars tympanica ossis temporalis*) [29]. Both parts are joined by the occipitotympanic suture (*sutura occipitotympanica*) that opens caudally into the jugular foramen (*foramen jugulare*). This foramen allows for the passage of, amongst others, the internal jugular vein, the glossopharyngeal nerve, the vagus nerve, and the accessory nerve (Figure 4C) [29].

The tympanic part of the temporal bone can now be further elaborated using Figure 4C. The tympanic bulla can be found directly lateral to the occipitotympanic suture. Lateral to the rostromedially projecting apex of the tympanic part is a short bony process. The internal carotid artery reaches the middle fossa of the cranial cavity (*fossa cranii media*) through the carotid canal (*canalis caroticus*), which is located immediately caudal to the tympanic bulla [29]. Only a fine osseous septum is present in between the internal carotid artery and the middle ear. Lateral to the carotid canal is the tubular osseous structure, called the external acoustic meatus (*meatus acusticus externus*). It is the osseous part of the ear canal. The external acoustic pore is the osseous ring that grants access to the external acoustic meatus.

The stylomastoid foramen (*foramen stylomastoideum*), located in between the insignificant mastoid process (*processus mastoideus*) and the styloid process (*processus styloideus*) of the petrous part of the temporal bone (*pars petrosa ossis temporalis*), leads to the facial canal (*canalis facialis*) (see Section 3.2.5 for further reading). The stylomastoid foramen is located caudal to the external acoustic meatus. In contrast, the petrosquamous foramen (*foramen petrosquamosum*) can be seen rostrally to this meatus, just caudal to the retroarticular process (Figure 4C). No *foramen lacerum* could be identified in *Papio hamadryas*.

#### 3.2.5. Cranial Cavity (*Cavum cranii*)

The cranial cavity encloses the brain. It is covered by the *calvaria*, which is the roof of the *neurocranium*. In the hamadryas baboon, the *calvaria* is formed by the frontal, parietal and temporal bones (Figure 4B). After the *calvaria* has been removed, the internal cranial base (*basis cranii interna*) can be inspected (Figure 6). It consists of three depressions, i.e., the rostral cranial fossa (*fossa cranii rostralis*), the middle cranial fossa (*fossa cranii media*) and the caudal cranial fossa (*fossa cranii caudalis*) (Figure 6B). These are elaborated below.

The rostral cranial fossa is located dorsally, immediately caudal to the orbits. It is bounded by the ethmoidal, sphenoidal, and frontal bones. These bones constitute the orbital sockets (Figure 3), which bulge into the rostral cranial fossa. In between these sockets sit a trough that is occupied by the paired olfactory bulb and paired olfactory tract [29]. The olfactory foramen (*foramen olfactorium*) is situated in the depth of the trough. Here, the bilateral olfactory nerve reaches the cranial cavity after numerous nervous filaments exiting the nasal cavity have perforated the cribriform plate of the ethmoidal bone [29]. The groove for the dorsal sagittal sinus (*sulcus sinus sagittalis dorsalis*, a dural sinus situated between the meningeal and periosteal layers of the *dura mater* [29]) can be observed dorsal to the olfactory foramen. The convergence of both edges of this sinus forms the frontal crest (*crista frontalis*), where the *falx cerebri* attaches (Figure 6A).

The caudal border of the rostral cranial fossa lies adjacent to the groove for the sphenoparietal sinus (*sulcus sinus sphenoparietalis*). This sinus originates near the junction of the sphenofrontal suture (*sutura sphenofrontalis*), and the suture between the wing of the presphenoid and the wing of the basisphenoid. From here, it runs in dorsolateral direction along the internal surface of the sphenoid bone, crossing the frontal bone towards the parietal bone. Towards the midline, the caudal border of the rostral cranial fossa courses from the origin of the sphenoparietal sinus (*sinus sphenoparietalis*) to the orbital foramen (Figure 4A and B). The middle cranial fossa is only partly formed by the frontal bone as it is primarily founded by the temporal and sphenoid bones. This fossa conceals a number of meaningful structures. The optic canal is positioned most rostrally in the middle cranial fossa, just paramedian to the chiasmatic groove (*sulcus chiasmatis*). It leads the optic nerve to the optic foramen in the orbital socket [29]. Ventrolateral to this opening, only separated by the clinoid process (*processus clinoideus*), the orbital foramen can be found. The oculomotorius nerve, the trochlear nerve, the ophthalmic nerve and the abducens nerve leave the cranial cavity by means of this foramen to innervate the orbital structures (Figure 6A). Caudal to the chiasmatic groove, flanked by a furrow in which the above-mentioned orbital nerves are situated, is a deeper depression termed the *sella turcica*. In the depth of the sella, the pituitary gland (*hypophysis*) is lodged in the hypophyseal fossa (*fossa hypophysialis*). The *sella turcica* is caudally delimited by the *dorsum sellae,* which presents two eminences, the paired caudal clinoid process (*processus clinoideus caudalis*). The *clivus* is the shallow depression caudal to the *dorsum sellae*. The sphenooccipital synchondrosis (*synchondrosis sphenooccipitalis*) can be appreciated here (Figure 6C,D). Caudolateral to the furrows for the orbital nerves lies the round foramen that leads to the ventral orbital fissure. It is the maxillary nerve that makes use of this aperture to reach the pterygopalatine fissure. The mandibular nerve travels through the oval foramen, which can be seen caudolateral to the round foramen. Medial to the oval foramen projects the tip of the petrosal part of the temporal bone (*apex partis petrosae*) in a rostromedial direction to the caudal clinoid processes. The petrooccipital fissure (*fissura petrooccipitalis*), the sphenopetrosal fissure (*fissura sphenopetrosa*), and the sphenooccipital synchondrosis meet at the apex of the petrosal part. The opening used by the internal carotid artery to reach the brain, i.e., the carotid canal, is to be found immediately lateral to this tip. The petrosquamous foramen is the tiny opening lateral to the petrous part of the temporal bone where it is at its maximal width. Caudal to this foramen, the groove for the petrosquamous sinus (*sulcus sinus petrosquamosi*) forms the apparent border between the petrosal and squamous parts of the temporal bone. Lateral to this foramen, channels for the greater and lesser petrosal nerves (*canalis n. petrosi majoris* and *canalis n. petrosi minoris*, respectively) can be discerned. The small petrosal nerve (from the glossopharyngeus nerve) enters the cranial cavity medial to the greater petrosal nerve (from the facial nerve) (Figure 6C,D) [29].

The caudal cranial fossa is almost entirely formed by the occipital bone. The lateral wall, however, is composed of the medial side of the petrous part of the temporal bone (Figure 6B). In this medial wall are two openings, i.e., the internal acoustic pore (*porus acusticus internus*) and the external aperture of the vestibular aqueduct (*apertura externa aqueductus vestibuli*) (Figure 6D). The facial nerve and the vestibulocochlear nerve exit the cranial cavity via the rostralmost internal acoustic meatus. The vestibulocochlear nerve enters the vestibular and cochlear mechanisms of the ear, whereas the facial nerve follows the route to the stylomastoid foramen, called the facial canal (*canalis facialis*), where it is accompanied by the stylomastoid artery [29]. The dorsal side or roof of the petrous portion of the temporal bone is termed the *tegmentum tympani*. Caudal to this tegmentum runs the prootic sinus in its groove (*sulcus sinus prootici*). It is continuous with the groove for the petrosquamous sinus (*sulcus sinus petrosquamosi*), which runs lateral to the petrous part. The jugular foramen is medial to the internal acoustic pore. It presents as a widening of the petrooccipital fissure. As mentioned earlier, this foramen gives passage to the internal jugular vein, the glossopharyngeal nerve, the vagus nerve, and the accessory nerve (Figure 4C). Medial to the jugular foramen, in the lateral wall of the large foramen, at the basis of the occipital condyle, two apertures appear. Both unite to form the *foramen nervi hypoglossi,* which allows for the hypoglossal nerve to leave the caudal cranial fossa [29]. The jugular tubercle (*tuberculum jugulare*) sits in between the jugular foramen and the *foramen nervi hypoglossi*. The impression of the cerebellar vermis (*vermis cerebellaris*), called *impressio vermialis*, is located directly caudal to the large foramen. The impressions of the bilateral cerebellar lobes (*fossae cerebellares*, singular: *fossa cerebellaris*) are located bilateral to the vermial impression. The sigmoidal groove (*sulcus sigmoideus*), which lodges part of the transverse sinus (*sinus transversus*), can be found medial to the petrosal part of the temporal bone, caudal to the jugular foramen. The transverse groove for the transverse sinus (*sulcus sinus transversi*) is caudolateral to the cerebellar vermis. The left and right transverse sinuses merge at the *confluens sinuum,* where the dorsal sagittal sinus is present.

### 3.3. Mandible

Compared to the skull, the mandible of the hamadryas baboon is substantial (Figure 7). It measured 13.7 cm and 14.1 cm in length in male skeleton 1 and male skeleton 2, respectively. The value for the female was 11.4 cm.

The left and right mandibles are rostrally joined by the synostotic mandibular symphysis (*symphysis mandibulae*). On a rostral view, this osseous concrescence leaves a small opening in the midline, i.e., the symphyseal foramen (*foramen symphysialis*). When viewed from caudal, multiple symphyseal foramina can be recognized (Figure 7D–F).

Each mandible consists of the rostral mandibular body (*corpus mandibulae*) and the caudal mandibular ramus (*ramus mandibulae*). The body is oriented horizontally, while the ramus takes a more vertical position by means of the mandibular angle (*angulus mandibulae*). A shallow indentation is present at the rostral side of the mandibular angle. Here, the facial artery and vein can be palpated at the level of the facial vascular notch (*incisura vasorum facialium*). The mandibular body holds the mandibular set of teeth. The incisors are present in the incisive part (*pars incisiva*) of the mandible, whereas the premolars and the molars can be observed in the molar part (*pars molaris*) (Figure 7A). As in the upper jaw, the incisors are two in number (the first incisor (*dens incisivus* (*mandibularis*) *primus*) and the second incisor (*dens incisivus* (*mandibularis*) *secundus*)), the single canine (*caninus*) sits on the angle of the alveolar arch (*arcus alveolaris*), the premolars are also two in number (the third premolar (*dens premolaris* (*mandibularis*) *tertius*) and the fourth premolar (*dens premolaris* (*mandibularis*) *quartus*)), and the molars are three in number (the first molar (*dens molaris* (*mandibularis*) *primus*), the second molar (*dens molaris* (*mandibularis*) *secundus*) and the third molar (*dens molaris* (*mandibularis*) *tertius*)) (Figure 7B,C). The diastema between the canine and the premolars is virtually non-existent (Figure 7C). All teeth are lodged in alveoli, which are arranged on the alveolar margin (*margo alveolaris*) of the mandibular body (Figure 7A). The opposite side of the mandibular body is the ventral margin (*margo ventralis*) (Figure 7A,D). The lateral surface of the mandibular body is known as the buccal surface (*facies buccalis*), whereas the surface that faces the tongue is the lingual surface (*facies lingualis*). Finally, the rostral surface of the alveolar arch is denoted as the labial surface (*facies labialis*) (Figure 7C). Multiple mental foramens (*foramina mentalia*) can be seen lateral to the symphyseal foramen, ventral to the root of the canine (Figure 7A,F).

The mental foramens are connected with the mandibular foramen (*foramen mandibulae*), which can be perceived at the medial side of the mandibular ramus ventral to the coronoid process (*processus coronoideus*), through the mandibular canal (*canalis mandibulae*) (Figure 7E). The *n.*, *a.* and *v. alveolaris ventralis* travel within this canal [29]. Ventral to the mandibular foramen is the groove where the mylohyoid muscle attaches (*sulcus mylohyoideus*) (Figure 7E) [29]. The coronoid process is a tin, triangular structure just rostral to the condylar process (*processus condylaris*). It has a convex rostral and a concave caudal side (Figure 7A). The temporal muscle inserts into the coronoid process [29]. The masseter muscle inserts more ventrally into the masseteric fossa (*fossa masseterica*), at the lateral side of the mandibular ramus (Figure 7B). At the medial side lies the pterygoid fossa (*fossa pterygoidea*), which is the insertion site of the medial pterygoid muscle (Figure 7A,C) [29]. In contrast, the pterygoid fovea (*fovea pterygoidea*), which serves as insertion site for the lateral pterygoid muscle, lies more dorsally at the rostromedial side of the condylar process (*processus condylaris*) (Figure 7C) [29]. The condylar process can be observed immediately caudal to the coronoid process. The mandibular notch (*incisura mandibulae*), where the *n. massetericus*, and the *a.* and *v. masseterica* run, is located in between both processes. The condylar process has a dorsal mandibular head (*caput mandibulae*), which is separated from the mandibular ramus by the mandibular neck (*collum mandibulae*) (Figure 7B). It plays a pivotal role in the formation of the temporomandibular joint (*articulatio temporomandibularis*), together with the mandibular fossa of the cranium. 

### 3.4. Hyoid Bone (Os hyoideum)

The hyoid apparatus (*apparatus hyoideus*) can shortly be referred to as the hyoid bone (Figure 8). It is rather large in *Papio*. Its location is ventromedial to the mandibular ramus. The term ‘hyoid apparatus’ refers to the fact that not a single ‘hyoid bone’, but five articulating bones are present. These include the unpaired body (*corpus*), and the paired lesser horn (*cornu minus*) and paired greater horn (*cornu majus*). In comparative anatomy of domestic mammals, the corpus is known as the *basihyoideum*, the *cornu minus* is termed *ceratohyoideum*, and the *cornu majus* is the *thyrohyoideum* [30]. The rostrodorsal aspect of the body is convex, whereas the caudal or pharyngeal side is excavated. In this hyoid bulla (*bulla hyoidea*) reside the laryngeal air sacs [29]. The large greater horn articulates at the caudodorsal portion of the body and projects in caudal direction. The tip of the greater horn shows a small tubercle. Dorsal to the articulation between body and greater horn sits the tiny lesser horn that projects dorsocaudally.

### 3.5. Vertebral Column

#### 3.5.1. General Conformation

The vertebral column is composed of five segments or regions, each consisting of a number of vertebrae (Figure 1). In the two male skeletons, the cervical region had seven cervical vertebrae, twelve thoracic vertebrae, seven lumbar segment vertebrae, three fused sacral vertebrae in the sacral region, and twenty caudal vertebrae in the caudal region or tail region. In the female skeleton, these numbers were the same except for twenty-one caudal vertebrae,

Besides the first two cervical vertebrae, the sacral vertebrae, and the most caudal vertebrae, each vertebra presents the same constitution. The vertebral body (*corpus vertebrae*) is cylindrical in shape. The cranial extremity is convex, whilst the caudal extremity (*extremitas caudalis*) is concave. The extremities of the consecutive vertebrae have indirect contact through the intervertebral disc (*discus intervertebralis*) [29]. A longitudinal median keel (*crista ventralis*) can be recognized at the ventral side of the vertebral body. It provides the attachment of the ventral longitudinal ligament of the vertebral column (*ligamentum longitudinale ventrale*) [29]. The vertebral arch (*arcus vertebrae*) sits on top of the body. This structure consists of the left and right fused laminae (singular: *lamina arcus vertebrae*), which connect to the body by means of the pedicles (singular: *pediculus arcus vertebrae*). The spinal process (*processus spinalis*) has its place on the dorsal junction of both laminae. A transverse process (*processus transversus*) projects bilaterally from the body or from the transition of the pedicle to the lamina. The vertebral arch, together with the dorsal side of the vertebral body, shape a large vertebral foramen (*foramen vertebrale*). The vertebral foramens (*foramina vertebralia*) of the consecutive vertebrae form the vertebral canal (*canalis vertebralis*) through which the spinal cord courses [29]. At the cranial and caudal sides of both pedicles project the bilateral cranial articular process (*processus articularis cranialis*) and the caudal articular process (*processus articularis caudalis*), respectively. The cranial and caudal articular processes of the consecutive vertebrae form synovial articulations. The articular facets of the cranial and caudal articular processes face dorsally and ventrally, respectively.

#### 3.5.2. Cervical Segment (*Vertebrae cervicales*)

The cervical segment of the vertebral column (Figure 9A) contains seven cervical vertebrae. It measured 8.3 cm and 8.9 cm in length in male skeleton 1 and male skeleton 2, respectively. The value for the female was 7.7 cm. The first cervical vertebra, i.e., the *atlas* and the second, i.e., the *axis*, are noticeably different from the other. These will therefore be discussed separately. The third to sixth cervical vertebrae are typical. The sixth cervical vertebra will be elaborated below as an example of a typical cervical vertebra. The seventh is an atypical cervical vertebra since it has characteristics in common with both a typical cervical and a thoracic vertebra. Its spinal process is prominent. The transverse process is heavy and long, resembling a short rib. In the hamadryas baboon, the transverse processes of the seventh cervical vertebra present the transverse foramen (*foramen transversarium*), as do all other cervical vertebrae. The caudal costal fovea (*fovea costalis caudalis*) is caudolaterally situated onto the vertebral body for articulation with the first rib.

##### Atlas

The *atlas* is the first vertebra of the vertebral column (Figure 9A). It supports the skull as it forms an articulation with the convex occipital condyles by means of its concave cranial articular foveae (singular: *fovea articularis cranialis*). The caudal articular foveae (singular: *fovea articularis caudalis*) are smaller and less concave. They articulate with the *axis*. A remarkable difference between the *atlas* and a typical vertebra is the lack of a vertebral body. Instead, the *atlas* presents a left and right lateral mass (*massa lateralis atlantis*), joined by the ventral and dorsal arches (*arcus ventralis atlantis* and *arcus dorsalis atlantis*, respectively) (Figure 9C,D). In the middle on the ventral and dorsal arches are small elevations that are identified as the ventral tubercle (*tuberculum ventrale*) and dorsal tubercle (*tuberculum dorsale*), respectively. The former is more pronounced than the latter as it is a hooklike caudal process. The dorsal tubercle is merely a rudiment of the spinous process. (Figure 9B–F). These tubercles serve as the sites of attachment for cervical muscles and ligaments [29]. The vertebral foramen is formed by both arches (Figure 9E,F). The concave, inner side of the ventral arch, presents an excavation for the initial vertebral body of the *atlas* that during evolution has been fused with the second cervical vertebra [29]. This excavation is the *fovea dentis* (Figure 9E). The transverse process sits laterally on the lateral mass (Figure 9C). Its basis is perforated by the transverse foramen (Figure 9F). This foramen is a unique characteristic of all cervical vertebrae. The cranial opening of the transverse foramen is dorsolaterally covered by a bony lamella of the dorsal arch. In addition, it is more dorsal than the caudal opening. As a result, the trajectory from the cranial to the caudal opening of the transverse foramen is dorsoventrally bent. The consecutive transverse foraminae create the transverse canal (*canalis transversarium*) through which the *a.* and *v. vertebralis* travel towards and from the brain, respectively [29,40]. The lateral vertebral foramen (*foramen vertebrale laterale*) is a foramen at the dorsolateral side of the dorsal arch, which allows for the first cervical nerve to leave the spinal cord [29,40]. Its medial opening lies within the cranial opening of the transverse foramen (Figure 9B,D). The alar foramen (*foramen alare*) can be found ventral to the lateral vertebral foramen, ventral to the transverse process (Figure 9B,C). Since this transverse process is wide in domestic mammals, it is termed the wing of the atlas (*ala atlantis*) in these species [41]. The foramen associated with this wing is then called the alar foramen. In the hamadryas baboon, this alar foramen also medially opens into the common cranial opening for both the transverse and lateral vertebral foraminae. 

##### Axis

The *axis* is the second cervical vertebra (Figure 10A). Its conformation is unique since the vertebral body of the *atlas* has been incorporated as the cranially projecting *dens axis* [29]. The dens shows an oval ventral surface for articulation (*facies articularis ventralis*) with the *fovea dentis* of the *atlas*. In addition, the dorsal articulating surface of the *dens* (*facies articularis dorsalis*) is traversed by the transverse ligament of the atlas (*ligamentum transversum atlantis*) (Figure 10B) [29]. The tip of the *dens* (*apex dentis*) is rather sharp (Figure 10B,C). Caudolateral to the *dens* lie the cranial articular facets (*facies articulares craniales,* singular: *facies articularis cranialis*) for articulation with the caudal articular foveae of the atlas (Figure 10B–D). Similar to the following cervical vertebrae, the *axis* presents with a vertebral body and vertebral arch that, together form the vertebral foramen (Figure 10D,E). The transverse processes are small caudolateral projections from the body and pedicle (Figure 10B–E). The transverse processes of each consecutive cervical vertebra gain length (Figure 10A). The transverse foramen is large and can be found at the base of the transverse process (Figure 10B,E). The spinous process is the high and wide dorsal extension from the vertebral arch (Figure 10B). Caudally, on each pedicle, is the caudal articular process (*processus articularis caudalis*) with the caudal articular facet (*facies articularis caudalis*) for articulation with the cranial articular process of the third cervical vertebra (Figure 10B–D).

##### Sixth cervical vertebra

The third to sixth cervical vertebrae have a comparable morphology. The lengths of the vertebral bodies are scant. In contrast, their widths are considerable. In between the vertebral arches is some space, called the interarcual space (*spatium interarcuale*). The consecutive vertebral foramens create the vertebral canal. The spinous processes increase in length, but lose some width (Figure 11A). The transverse processes present the transverse foramen at their bases. They end in two tubercles, a dorsal and a ventral (*tuberculum dorsale* and *tuberculum ventrale*, respectively) (Figure 11B,C). The ventral tubercle of the transverse process of the sixth cervical vertebra is wide. It can therefore be termed the ventral lamina (*lamina ventralis*) (Figure 11A). A groove is formed in between both tubercles. It is in this *sulcus nervi spinalis* that the cervical nerves leave the spinal cord [29]. The sixth cervical vertebra is here discussed as a typical cervical vertebra. However, this vertebra has many features in common with the *axis*. As mentioned earlier, the seventh cervical vertebra presents an articular facet for the first rib (*fovea costalis caudalis*) (Figure 11A).

#### 3.5.3. Thoracic Segment (*Vertebrae thoracicae*)

The thoracic segments of the two male and the one female baboon skeletons comprised twelve thoracic vertebrae. The thoracic segment of male skeleton 1 measured 17.4. cm in length, and that of male skeleton 2 measured 18.1 cm in length. The value for the female was 15.1 cm. The dorsoventral dimension of the vertebral body increases from the first to the twelfth vertebra (Figure 12A). The vertebral foramens are smaller compared to those of the cervical vertebrae. The thoracic vertebrae are unique in that they present facets for articulation with the ribs. The number of rib pairs equals the number of thoracic vertebrae. Another typical trait is the pronounced spinous process.

Articulation with the head of the ribs is accommodated by the bilateral presence of an articular facet at the craniolateral as well as at the caudolateral side of the vertebral body. These facets are termed *fovea costalis cranialis* and *fovea costalis caudalis*, respectively (Figure 12A). More specifically, each rib articulates with two consecutive vertebrae. Indeed, the head of the rib fits within the cavity formed by the caudal costal fovea of the precedent vertebra and the cranial costal fovea of the subsequent vertebra. The head of the first rib articulates with the caudal costal fovea of the seventh cervical vertebra and the cranial costal fovea of the first thoracic vertebra. In addition, a second synovial joint is formed between the costal tubercle (*tuberculum costae*) and the costal fovea on the transverse process (*fovea costalis processus transversi*) of the corresponding vertebra (Figure 12A). Thus, the tubercle of the first rib articulates with the transverse process of the first thoracic vertebra, etc.; however, the horizontally projecting transverse processes decrease in length and volume towards the lumbar region (Figure 12A–C). Moreover, the last three caudal thoracic vertebrae lack the articular facet on their transverse processes (Figure 12A). As a result, the articulations with the ribs simplify allowing for increased costal mobility.

The spinous processes are large (Figure 12B,C). In the first three thoracic vertebrae, they become somewhat less tall and incline more and more in caudal direction. Then, the spinous processes grow until the ninth vertebra to decrease in length again towards the lumbar region. They additionally erect such that the spinous process of the eleventh thoracic vertebra presents a cranial inclination instead of a caudal. The eleventh thoracic vertebra is therefore the anticlinal vertebra (*vertebra anticlinalis*) (Figure 12A). 

On a lateral view, a small notch can be observed at the cranial side of the pedicle (*incisura vertebralis cranialis*), whereas a larger notch is seen at the caudal side of the pedicle (*incisura vertebralis caudalis*). When two consecutive vertebrae are considered, an intervertebral foramen (*foramen intervertebrale*) is created in between both notches (Figure 12A). This foramen enables the thoracic nerve to leave the spinal cord [29]. As in the cervical segment, the articular processes are almost horizontally oriented (Figure 12B,C). 

#### 3.5.4. Lumbar Segment (*Vertebrae lumbales*)

The lumbar segment of male skeleton 1 measured 15.7. cm in length, and that of male skeleton 2 measured 16.3 cm in length. The value for the female was 17.3 cm. The lumbar vertebrae, which were seven in number in the three examined baboon skeletons, are massive (Figure 13A). Consequently, the lumber region is, despite the reduced number of vertebrae, not much shorter in comparison with the thoracic segment. Like the bodies, the pedicles and laminae are weighty. The vertebral foramens are larger than those of the thoracic vertebrae.

A very remarkable trait is the heavy, quadrilateral spinous process of each lumbar vertebra, not in the least of those vertebrae in the middle of the lumbar region (Figure 13A). In addition, the transverse processes increase both in length and in width towards the fifth lumbar vertebra. Those of the sixth and seventh lumbar vertebrae are less pronounced and slightly bend in a cranial direction. However, the right transverse process of the first lumbar vertebra was long in one of the examined specimens. It resembled a short rib (Figure 13B).

The cranial and caudal articular processes are also prominent. It should be noticed that the articular facets are vertically oriented (Figure 13C,D). Compared to the caudal articular process, the cranial is even more protuberant on a lateral view (Figure 13A). This is due to the presence of the mamillary process (*processus mamillaris*) (Figure 13A,C). Just ventral to the caudal articular process sits the caudally projecting accessory process (*processus accessorius*) in the first five lumbar vertebrae (Figure 13A). Finally, it can be perceived that the interarcual spaces are wide in the lumbar region (Figure 13B).

#### 3.5.5. Sacrum, *Os sacrum* (*Vertebrae sacrales*)

The triangular sacrum is composed of a number of fused sacral vertebrae. It measured 5.2. cm in length in male skeleton 1, and 5.5 cm in male skeleton 2. The value for the female was 5.4 cm. In the three specimens that were examined in the present study, this number was three. As such, a solid basis is created for articulation with the pelvic bones. The cranial side of the cranial-most fused sacral vertebra forms the basis of the sacrum (*basis ossis sacri*) (Figure 14A,C,D). The ventral rim of the body of the first sacral vertebra is the promontory (*promontorium*) that forms the roof of the entrance of the pelvic cavity (Figure 14A,D). The caudal apex of the sacrum (*apex ossis sacri*) is represented by the caudal aspect of the third fused sacral vertebra (Figure 14A,B). The ventral side of the sacrum, which faces the pelvic cavity (*cavum pelvis*), hence the term pelvic face (*facies pelvina*), shows a smooth, vaguely concave surface (Figure 14D). The dorsal side (*facies dorsalis*) is somewhat convex and presents multiple typical structures. The median sacral crest (*crista sacralis mediana*) arises from the fusion of the spinous processes of the three fused sacral vertebrae (Figure 14A–C). The origin of the intermediate sacral crest (*crista sacralis intermedia*) is the fusion of the articular processes (Figure 14C). Finally, the united transverse processes craft the lateral sacral crest (*crista sacralis lateralis*). Although fused, all individual processes can still be recognized (Figure 14A–C). In particular, the transverse processes of the first sacral vertebra are very pronounced. This bilateral sacral wing (*ala sacralis*) presents a dorsolaterally facing ear-shaped facet (*facies auricularis*) for articulation with the hip bone (Figure 14A–D). Caudomedial to the auricular surface lies the sacral tuberosity (*tuberositas sacralis*) (Figure 14B,C).

The two fusion sites of the bodies of the three sacral vertebrae can be recognized on the pelvic surface of the sacrum by the transverse lines (*lineae transversae*) (Figure 14D). Bilateral to these lines remain openings at both the dorsal and ventral sides of the sacrum (*foramen sacralis dorsalis* and *foramen sacralis ventralis*) (Figure 14C,D). Each of the three vertebral foramens form, in analogy with the vertebral canal, the sacral canal (*canalis sacralis*) (Figure 14A,B). Sacral nerves and the meninges run within this canal [29].

#### 3.5.6. Caudal Segment (*Vertebrae caudales* or *Vertebrae coccygeae*)

The length of the caudal segment of male skeleton 1 was 43.7 cm, and 45.7 cm in male skeleton 2. The value for the female was 39.1 cm. The caudal vertebrae represent the last, most caudal segment of the vertebral column. Both male individuals that were examined in the present study had twenty caudal vertebrae. The female skeleton had twenty-one caudal vertebrae. The more cranial caudal vertebrae resemble a typical vertebra. They have short bodies, cranial and caudal articular processes sitting on the vertebral arch that forms the vertebral foramen together with the body, transverse processes, and spinous processes. Intervertebral foramens can be identified on a lateral view. The vertebral bodies elongate first until the seventh caudal vertebra to shorten again towards the tip of the tail. In addition, the bodies’ diameters shrink, and the processes fade to disappear ultimately. This is visualized in Figure 15.

### 3.6. Bony thorax (Skeleton thoracis)

#### 3.6.1. Ribs (*Costae*)

A rib (*costa*) is composed of the bony rib (*os costale*) and the costal cartilage (*cartilago costalis*). Together with the twelve thoracic vertebrae, which have been described above, and the sternum, which is described in the next paragraph, the twelve pairs of ribs form the bony thorax (Figure 1). The length of the bony ribs increases until rib 9. Ribs 10, 11 and 12 become consistently shorter (Figure 16A).

The bony ribs are composed of a head (*caput costae*), a neck (*collum costae*) and a body (*corpus costae*). The transition between the neck and the body is sharply bent. This is the costal angle (*angulus costae*). The lateral sides of the ribs are convex and have a smooth surface, whereas the medial sides are rather concave, which is owed by the longitudinal groove (*sulcus costae*) in which intercostal nerves and blood vessels run (Figure 16B) [29].

As explained during the discussion of the thoracic vertebrae, the heads of the ribs articulate with the caudal and cranial costal foveae of two consecutive vertebrae. An additional articulation is present between the costal tubercle and the costal fovea of the transverse process of the corresponding thoracic vertebra. To this purpose, the head and the tubercle of the rib are equipped with articular facets, i.e., *facies articularis capitis costae* and *facies articularis tuberculi costae*, respectively (Figure 16B). However, rib after rib, the head and the tubercle are positioned closer to each other. In other words, the neck of the rib becomes shorter (Figure 16A,B). In well-developed necks, a dorsal crest (*crista colli costae*) can be identified (Figure 16B). The last three ribs fail to present the articular facet on the tubercle, if present. Indeed, the eleventh rib almost lacks the tubercle. It is completely absent in the last rib (Figure 16A). As mentioned during the discussion of the thoracic vertebrae, the articulations between the ribs and the thoracic vertebrae simplify towards the lumbar region.

The distal extremity of each bony rib is provided by a costochondral junction (*articulatio costochondralis*) where the costal cartilage attaches. Each rib, except the last, achieves direct or indirect contact with the sternum by means of its costal cartilage (Figure 1). Ribs 1 to 8 realize a direct contact with the sternum through their costal cartilages. These ribs are true ribs (*costae verae/sternales*, singular: *costa vera/sternalis*). Ribs 9 and 10 only have indirect contact with the sternum since their costal cartilages join the costal cartilage of the previous rib. In this way, the costal arch (*arcus costalis*) is formed. Such ribs are false ribs (*costae spuriae/asternales*, singular: *costa spuria/asternalis*). Finally, ribs 11 and 12 have no contact with the sternum. These asternal ribs are floating (*costae fluctuantes*, singular: *costa fluctuans*) (Figure 16A).

#### 3.6.2. Sternum

The sternum is placed at the ventral side of the thoracic cage (Figure 1). This structure was 13.8 cm long in male skeleton 1, and 14.7 cm long in male skeleton 2. The value for the female was 13.5 cm. It was composed of seven segments in the three examined specimens. The cranial-most segment, i.e., the sternal manubrium (*manubrium sterni*), is triangular in shape. Its cranial basis presents a median indentation termed the jugular notch (*incisura jugularis*). Bilateral to this structure lies the clavicular notch (*incisura clavicularis*) for articulation with the clavicle. The costal cartilages of the first pair of ribs articulates with the costal notches (singular: *incisura costalis*) at the lateral aspects of the *manubrium*. The caudally oriented *apex* of the triangular *manubrium* articulates with the first *sternebra* by means of the cartilaginous manubriosternal synchondrosis (*synchondrosis manubriosternalis*). Bilateral to this synchondrosis is the costal notch for the costal cartilages of the second pair of ribs. There are five sternebrae in total, which are joined by sternal synchondroses (singular: *synchondrosis sternalis*). These sternebrae constitute the sternal body (*corpus sterni*). The costal cartilages of the subsequent pairs of ribs articulate bilaterally to these synchondroses. The caudal-most sternal segment is the xiphoid process (*processus xiphoideus*), which is caudally elongated by the xiphoidal cartilage (*cartilago xiphoidea*). The lateral aspects of the synchondrosis between the fifth sternebra and the xiphoid process present costal notches for articulation with both the seventh and eighth costal cartilages. Figure 17 is a visual representation of the here-described sternum. 

### 3.7. Thoracic Limb

#### 3.7.1. Collar Bone, Clavicle (*Clavicula*)

The collar bone, which is depicted in Figure 18, is sigmoidally shaped. Its length was 6.7 cm in male skeleton 1, and 7.0 cm in male skeleton 2. The value for the female was 5.7 cm. It extends from the clavicular notch on the *manubrium* of the sternum to the *acromion* of the shoulder blade. Its medial extremity is therefore called the *extremitas sternalis*, while its lateral extremity is the *extremitas acromialis*. Both extremities have their articular surfaces. These are the sternal articular surface (*facies articularis sternalis*) and the acromial articular surface (*facies articularis acromialis*). The remainder in between both extremities is the body of the shoulder blade (*corpus claviculae*). The conoid tubercle (*tuberculum conoideum*) is faint in the hamadryas baboon. This tubercle allows for the conoid ligament, which is part of the coracoclavicular ligament, to attach [29]. The trapezoid line (*linea trapezoidea*) sits dorsolateral to the conoid tubercle. The groove for the subclavius muscle is located more medially. At the sternal extremity, the shallow impression for the costoclavicular ligament (*impressio ligamenti costoclavicularis*) can be identified.

#### 3.7.2. Shoulder Blade (*Scapula*)

The shoulder blade is the larger bone of the pectoral girdle. As this bone has a typical triangular shape, it presents a cranial angle (*angulus cranialis*), a caudal angle (*angulus caudalis*), and a ventral angle (*angulus ventralis*). The width between the cranial and caudal angles was 8.7 cm in male skeleton 1, and 9.2 cm in male skeleton 2. The value for the female was 7.5 cm. The dorsal margin (*margo dorsalis*) connects the cranial with the caudal angle and is directed towards the back of the animal. The cranial margin (*margo cranialis*) runs from the cranial to the ventral angle. The caudal margin (*margo caudalis*) lies at the opposite side, in between the caudal and ventral angles. Both margins show a clear convergence at the level of the scapular neck (*collum capulae*). In particular, the cranial margin sharply bends in a caudal direction at the scapular notch (*incisura scapulae*) (Figure 19A,B). The scapular spine (*spina scapulae*) is the very characteristic structure at the lateral side of the scapula, which extends from the dorsal margin to the ventral angle, hereby dividing the lateral side into a narrow supraspinous fossa (*fossa supraspinata*) and a much wider infraspinous fossa (*fossa infraspinata*). The eponymous muscles, i.e., supraspinous muscle (*musculus supraspinatus*) and infraspinous muscle (*musculus infraspinatus*) have their origins here. The spine becomes more robust towards the ventral angle, detaches from the shoulder blade, and slightly curves in a cranial direction to ultimately bend caudally. This is the *acromion*, which connects with the acromial extremity of the collar bone (Figure 19A). The length of the shoulder blade from the dorsal margin to the tip of the acromion was 13.2 cm in male skeleton 1, and 13.8 cm in male skeleton 2. The value for the female was 11.9 cm. Both the scapular spine and the acromion are sites of origin for the deltoid muscle and trapezius muscle [29]. The acromial part of the deltoid muscle has its origin on the acromion. The spinal part finds its origin at the level of the scapular spine [29]. 

At the ventral angle of the scapula, the concave glenoid cavity (*cavitas glenoidalis*) can be noticed (Figure 19A,B,E). Here, articulation with the humeral head is achieved. The caudal aspect of the scapular neck presents the site of origin of the lesser teres muscle, i.e., the infraglenoid tubercle (*tuberculum infraglenoidale*) (Figure 19A,B,D) [29]. At the opposite side of the neck, the supraglenoid tubercle (*tuberculum supraglenoidale*) allows for the long head of the biceps muscle to originate (Figure 19C,E) [29]. In between the supraglenoid tubercle and the glenoid cavity sits the glenoid notch (*incisura glenoidalis*) (Figure 19E). At the craniomedial side of the scapular neck, the coracoid process, which is the origin of the short head of the biceps muscle, the coracobrachial muscle, and the lesser pectoral muscle, bends sharply in a caudomedial direction (Figure 19B–E) [29].

The medial side of the shoulder blade is characterized by the slightly concave subscapular fossa (*fossa subscapularis*). Actually, it presents a cranial, a middle, and a caudal excavation. The subscapular muscle originates here [29]. The attachment of the ventral serratus muscle is marked by the rough surfaces near the cranial and caudal angles of the shoulder blade (Figure 19B) [29].

#### 3.7.3. Skeleton Brachii: *Humerus*

The stylopodium of the thoracic limb consists of the humerus. Together with the distal aspect of the *scapula*, the proximal aspect of this bone forms the shoulder joint (*articulatio humeri*). The dorsocaudally oriented humeral head (*caput humeri*) articulates with the glenoid cavity of the shoulder blade (Figure 20A,C,D). The humeral neck (*collum humeri*) forms the transition between the head and the shaft of the humerus (*corpus humeri*) (Figure 20A, C and D). The length of the humerus from its most proximal to its most distal extremity was 21.3 cm in male skeleton 1, and 22.7 cm in male skeleton 2. The value for the female was 17.8 cm.

The greater tuberosity (*tuberculum majus*) rises craniolateral to the humeral head (Figure 20A,B,D). The infraspinatus muscle inserts on the smooth surface at the lateral side of the greater tuberosity (*facies musculi infraspinati*) (Figure 20A) [29]. At the opposite side sits the lesser tuberosity (*tuberculum minus*), craniomedial to the humeral head (Figure 20B,C). The tendon of the long head of the biceps brachii muscle fits in between both tuberosities, which are widely separated from each other by the intertubercular groove (*sulcus intertubercularis*) (Figure 20B,C). The lateral and medial lips of this groove are formed by the distal twig of the cranial portion of the greater and lesser tuberosity, respectively. The lateral lip of the groove, i.e., the crest of the greater tuberosity (*crista tuberculi majoris*), forms the cranial margin of the proximal third of the shaft of the humerus (Figure 20A–C). The distal twig of the caudal portion of the greater tuberosity serves as the site of origin for the lateral head of the triceps brachii muscle [29], hence its name *linea musculi tricipitis* (Figure 20A,D). The distal twig of the caudal portion of the lesser tuberosity presents an elongated prominence, where the teres major and latissimus dorsi muscles insert, termed the *tuberositas teres major* (Figure 20C) [29]. The deltoid tuberosity (*tuberositas deltoidea*) can be identified on the lateral side of the shaft (*corpus humeri facies lateralis*), at the transition from its proximal to its middle-third, caudolateral to the cessation of the crest of the greater tuberosity (Figure 20A,B). The name of this structure refers to the deltoideus muscle that inserts here [29].

The shaft presents a cranial convexity (Figure 20A,C). The bending is emphasized by the pronounced cessation of the crest of the greater tuberosity, which is a prominent feature on the cranial side of the shaft (*corpus humeri facies cranialis*) (Figure 20B). The medial side of the shaft (*corpus humeri facies medialis*) has a level surface, except for the tuberosity of the teres major muscle (Figure 20C). The caudal side of the shaft *(corpus humeri facies caudalis*) is devoid of any prominent structure (Figure 20D).

The distal aspect of the humerus is easily identified by its condyle (*condylus humeri*) (Figure 20A,C). This cylindrical structure possesses a smaller lateral and a larger medial protuberance known as the lateral epicondyle (*epicondylus lateralis*) and the medial epicondyle (*epicondylus medialis*), respectively (Figure 20A–D). These epicondyles permit the attachment of the respective collateral ligaments. A sharp ridge leaves the lateral epicondyle in proximal direction to quench at the caudal side of the humerus at the transition from the middle to the distal third of the shaft. This lateral supracondylar ridge (*crista supracondylaris lateralis*) is the site of origin of the supinator muscle, hence the alternative name *crista supinatoris*, and the extensor musculature of the wrist and hand (Figure 20A,B) [29]. A comparable ridge is seen at the medial side. However, this medial supracondylar ridge (*crista supracondylaris medialis*) is much less pronounced (Figure 20B,C). The pronator muscle originates here [29].

The articular surface at the medial aspect of the condyle that articulates with the trochlear notch of the *ulna* is the *trochlea humeri* (Figure 20B). The coronoid fossa can be observed proximal to the *trochlea* at the cranial side of the humerus (Figure 20B). The ulnar coronoid process fits here when the elbow joint is bent. The *trochlea* is not well demarcated from the lateral *capitulum* (Figure 20B). This structure articulates with the head of the *radius*. In analogy with the coronoid fossa, the radial fossa (*fossa radialis*) is the depression proximal to the *capitulum* where the radial head accommodates when the elbow joint is flexed (Figure 20B). A much deeper excavation can be seen at the caudal side of the humerus, immediately proximal to the condyle. This *fossa olecrani* allows for receiving the *olecranon* of the *ulna* when the elbow is in flexion (Figure 20D).

#### 3.7.4. Skeleton Antebrachii: *Radius* and *Ulna*

The antebrachial skeleton consists of the *radius* and the *ulna*. It is slightly wrung since the proximal end of the *radius* articulates with the humeral *capitulum*, which is located at the lateral side of the humeral condyle, and the distal end articulates with the medial aspect of the wrist. When the *ulna* is considered, it can be noticed that the proximal extremity articulates with the humeral *trochlea*, which is located at the medial side of the humeral condyle, and the distal extremity articulates with the lateral aspect of the wrist. At the level of the elbow joint, the *radius* and *ulna* are, however, not positioned next to each other but in front of each other with the *radius* cranial to the *ulna*. In contrast, the *radius* is positioned medial to the *ulna* at the level of the carpal joint. The torsion in the antebrachial skeleton can be amplified by endorotation (pronation) or decreased by exorotation (supination) (Figure 1).

##### Radius

The length of the radius from its most proximal to its most distal extremity was 18.4 cm in male skeleton 1, and 20.1 cm in male skeleton 2. The value for the female was 17.9 cm. The *radius* presents a lateral curvature in its body (*corpus radii*). This can certainly be appreciated when the *radius* is viewed cranially, with the cranial surface of the radial body (*corpus radii facies cranialis*) in sight, or on a caudal view of the *radius*, thus observing the caudal surface of the radial body (*corpus radii facies caudalis*) (Figure 21A). The round head of the *radius* (*caput radii*) is delimited from the body by means of the smaller, cylindrical neck (*collum radii*) (Figure 21A–C). The vaguely concave proximal surface of the radial head (*fovea articularis*) articulates with the *capitulum* of the humeral condyle. Since the radial head is higher laterally than medially, the articular surface is oblique (Figure 21C). This characteristic allows for the *radius* to deviate towards the medial plane, resulting in the distal extremity sitting medial to the *ulna*. The smooth caudal rim of the radial head articulates with the radial notch of the *ulna*. This is the articular circumference (*circumferentia articularis*) (Figure 21B).

The radial tuberosity (*tuberositas radii*), an elongated protuberance at the caudomedial side of the radial shaft, directly distal to the radial neck, is the insertion site of the biceps brachii muscle and the medial collateral ligament of the elbow joint (Figure 21B,C) [29]. An oblique line (*linea obliqua*) originates from the distal tip of this tuberosity (Figure 21C). The lateral margin of the radial shaft (*corpus radii margo lateralis*) is sharper than the medial margin (*corpus radii margo medialis*). The antebrachial interosseous membrane (*membrana interossea antebrachii*) attaches here [29]. The lateral margin is therefore sometimes called the interosseous margin (*margo interosseus*) [29].

The distal extremity of the *radius* is the widest segment of the *radius*. This radial trochlea (*trochlea radii*) is characterized by the medial radial styloid process (*processus styloideus radii*), which is a conical bony projection (Figure 21A–C). Several structures, including the brachioradialis muscle, the abductor pollicis longus muscle, and the lateral collateral ligament of the wrist, attach here [29]. At the opposite, thus lateral side, the ulnar notch is an irregular indentation in which the styloid process of the ulna lodges (Figure 21A,B). At the caudal side, the rough transverse crest (*crista transversa*) serves as the site of origin of several short flexor muscles and ligamentous structures such as the joint capsule and the *retinaculum flexorum* (Figure 21C) [29].

##### Ulna

The *ulna* is a long, tapered bone with a very characteristic proximal extremity, i.e., the *olecranon*, and a pointed distal end, i.e., the styloid process (*processus styloideus ulnae*). The length of the radius, measured between both mentioned structures was 21.7 cm in male skeleton 1, and 22.6 cm in male skeleton 2. The value for the female was 19.3 cm. The apex of the olecranon is reinforced as *tuber olecrani* (Figure 22A–C). This is the insertion site of the triceps muscle [29]. The humeral *trochlea* articulates within the trochlear notch (*incisura trochlearis*) (Figure 22A–C). It presents a proximal process that is directed cranially. This anconean process (*processus anconeus*) penetrates the *fossa olecrani* of the humerus when the elbow joint is extended (Figure 22B,C). In addition, the trochlear notch possesses a distal, medially oriented coronoid process (*processus coronoideus*), which serves as insertion site for the brachialis muscle and a branch of the medial collateral ligament (Figure 22A,C). The ulnar tuberosity (*tuberositas ulnae*), where a branch of the brachialis muscle inserts, can be found caudomedially to this process (Figure 22C). The radial notch (*incisura radialis*), where the articular circumference of the radial head articulates, is located lateral and somewhat distal to the coronoid process (Figure 22A,B).

The body of the *ulna* (*corpus ulnae*) is triangular and therefore presents a cranial, a lateral, and a medial surface (*facies cranialis*, *facies lateralis*, and *facies medialis*, respectively) (Figure 22A–C). The cranial surface gets sharper in distal direction and becomes the interosseous margin (*margo interosseus*). The antebrachial interosseous membrane attaches here [29]. The three surfaces of the ulnar body are bounded by the lateral, the caudal, and the medial margins (*margo lateralis*, *margo caudalis*, and *margo medialis*, respectively) (Figure 22A–C).

The distal extremity or head of the *ulna* (*caput ulnae*) is very characteristic. The pointed styloid process is directed towards the wrist (Figure 22A–C). Its craniomedial side articulates with the ulnar carpal bone or *os triquetrum* through a small articular surface (*facies articularis carpea*) (Figure 22A,C). The caudomedial articular surface for articulation with the ulnar notch of the *radius* (*circumferentia articularis*) is slightly larger (Figure 22C).

#### 3.7.5. Skeleton Manus

The skeleton of the hand (*skeleton manus*) is complex since it possesses over 35 bones and sesamoid bones. Regarding the bones, it is composed of the carpal bones, the five metacarpal bones, and the bones within the five fingers (*ossa digitorum manus*) (Figure 23). The length was 11.7 cm in male skeleton 1, and 12.6 cm in male skeleton 2, measured to the tip of the third digit. The value for the female was 9.2 cm.

In total, there are nine carpal bones that form the *carpus*. They are organized into the proximal or antebrachial row, which articulates with the *radius* and *ulna*, and the distal or metacarpal row that articulates with the metacarpal bones. The central carpal bone (*os carpi centrale*) sits in between both rows. From medial to lateral, the antebrachial row is composed of the scaphoid (*os scaphoideum* or *os carpi radiale*), the lunate (*os lunatum* or *os carpi intermedium*), the triquetrum (*os triquetrum* or *os carpi ulnare*), and the pisiform (*os pisiforme* or *os carpi accessorium*). The metacarpal row presents, also from medial to lateral, the trapezium (*os trapezium* or *os carpale primum*), the trapezoid (*os trapezoideum* or *os carpale secundum*), the capitate (*os capitatum* or *os carpale tertium*) and the hamate (*os hamatum* or *os carpale quartum*). The central carpal bone articulates with the scaphoid and lunate of the proximal row, and with the trapezium, trapezoid, and capitate of the distal row. The sesamoid bone, which is present in the long abductor muscle of the thumb (*os sesamoideum m. abductoris digiti primi / pollicis longi*), can be observed laterally to the scaphoid after incomplete maceration. It should not be mistaken for a carpal bone (Figure 23B).

The five metarcarpal bones are slender and have broadened proximal and distal extremities. The body (*corpus*) of each metacarpal has a dorsal surface (*facies dorsalis*), and a palmar surface (*facies palmaris*). It presents an oval transection as the body is flattened in dorsopalmar direction. As a consequence, the lateral margin (*margo lateralis*) and the medial margin (*margo medialis*) are more acute. The proximal basis is somewhat concave as it forms an articulation with the metacarpal carpal bones by means of its articular surface (*facies articularis*). The medial-most or first metacarpal bone (*os metacarpale primum*) articulates with the trapezium, the second metacarpal bone (*os metacarpale secundum*) articulates with the trapezoid and the capitate, the third metacarpal bone (*os metacarpale tertium*) articulates with the capitate, the fourth metacarpal bone (*os metacarpale quartum*) articulates with the capitate and hamate, and the lateral-most metacarpal bone (*os metacarpale quintum*) articulates with the hamate. The distal head (*caput*) of each metacarpal is characterized by a rounded *trochlea* that forms a joint with the respective digit. The palmar side of each metacarpal *trochlea* bears a pair of oval sesamoid bones (*ossa sesamoidea palmaria*) (Figure 23C).

There are five fingers or digits (*digiti*, singular: *digitus*). Their supporting bones include three phalanges, except the thumb, which has only two. The three phalanges are the proximal, middle, and distal phalanx (*phalanx proximalis*, *phalanx media* and *phalanx distalis*, respectively). The phalanges of the thumb (*phalanges pollicis*) comprise the proximal phalanx and the distal phalanx. In analogy with the metacarpal bones, the phalanges own a proximal base (*basis*), a body (*corpus*), and a distal head (*caput*). The articular fovea (*fovea articularis*) on the base of each phalanx articulates with the *trochlea* of the bone that is located proximally to it. This might thus either be a metacarpal, a proximal phalanx, or a middle phalanx. The distal phalanx can easily be differentiated from the other phalanges as its body tapers to almost come to a point. However, the tip itself is bulged by a tuberosity (*tuberositas phalangis distalis*). The distal phalanx of the thumb is broader than that of the other digits and its tuberosity is more robust.

When the total lengths of the digits are compared, the following digital formula can be deduced: III > IV > II > V > I. The *pollex* is the shorter of the five digits, not only because of the absence of the middle phalanx, but also because of the shorter metacarpal, and proximal and distal phalanges.

### 3.8. Pelvic Limb

#### 3.8.1. Hip Bone (*Os coxae*)

The left and right pelvic girdles are composed of the left and right *ossa coxae*. Each *os coxae* is formed by the junction of the ilium, ischium, and pubis (Figure 24A). They join in the *acetabulum*. This deeply excavated cup forms the hip joint (*articulatio coxae*), together with the femoral head. More specifically, the lunar articular surface (*facies lunata*) is covered with articular cartilage. This surface is crescent, which means that the acetabular margin (*margo acetabularis*) is discontinuous. Indeed, a ventral acetabular notch (*incisura acetabuli*) can be identified (Figure 24A).

The ilium is located craniodorsally in the *hip bone*. It consists of two parts. The caudal body (*corpus ossis ilii*) is positioned in the center of the hip bone where it connects with the ischium and pubis (Figure 24C). The long, cranial wing (*ala ossis ilii*) forms the sacroiliac joint (*articulatio sacroiliaca*) by means of the auricular surface (*facies auricularis*), which is placed on its ventromedial side (Figure 24B and C). This ventromedial side of the wing that is directed towards the sacrum is the *facies sacropelvina*. The iliac tuberosity (*tuberositas iliaca*), for the attachment of ligaments and short muscles, can be noticed cranially to the auricular surface (Figure 24C) [29]. The iliac fossa (*fossa iliaca*) is the smooth, faintly excavated surface positioned ventrolateral to this structure, where the iliopsoas muscle inserts (Figure 24C) [26]. In contrast, the lateral surface of the wing is clearly excavated. It is the site of origin of the gluteus medius muscle [29], hence the name gluteal surface (*facies glutea*) (Figure 24A,B). This excavation is bordered by the cranial iliac crest (*crista iliaca*), the dorsocranial iliac spine (*spina iliaca dorsalis cranialis*), which serves as an attachment site of a sacroiliac ligament and short muscles, the dorsocaudal iliac spine (*spina iliaca dorsalis caudalis*), the ventrocranial iliac spine (*spina iliaca ventralis cranialis*), where the sartorius and tensor fasciae latae muscles originate, and the ventrocaudal iliac spine (*spina iliaca ventralis caudalis*), with the sites of origin of the rectus femoris muscle (*area musculi recti femoris*) and the lesser psoas muscle (*tuberculum musculi psoas minoris*) (Figure 24A,B) [29].

The ischium is positioned caudally in the hip bone. A very pertinent structure of this bone is the ischial tuberosity (*tuber ischiadicum*). Quite a few muscles (amongst others adductor and hamstrings muscles) find their origins here [29]. This tuberosity presents a large, oval, and rough surface that supports the overlying ischial callosities [29]. The distance between the acetabulum and the ischial tuberosity is rather large, which is due to the elongation of the ischial body (*corpus ossis ischii*). Towards the ischial tuberosity, the ischial body widens to form a slightly concave surface, i.e., the ischial table (*tabula ossis ischii*). From this table, the branch of the ischium (*ramus ossis ischii*) extends medially (Figure 24C). The ischial spine (*spina ischiadica*) is located dorsally to the acetabulum and projects dorsomedially. Together with the dorsocaudal iliac spine, which is positioned cranially to the ischial spine, a large sciatic notch (*incisura ischiadica major*) is formed as an elongated depression along the dorsal border of the ilium (Figure 24A,B). The lesser sciatic notch (*incisura ischiadica minor*) is only half the length of the greater. It is shaped in between the ischial spine and the ischial tuberosity (Figure 24A–C).

The pubis can be found ventrally in the hip bone. Its cranial branch (*ramus cranialis ossis pubis*) extends in a medial direction to unite with the ilium, whereas the caudal branch (*ramus caudalis ossis pubis*) is directed caudally. As such, it makes contact with the branch of the ischium (Figure 24C). Several adductor muscles originate at the ventral aspect of both branches [29]. The obturator foramen (*foramen obturatum*) is the opening that is left by the cranial and caudal branches of the pubis, and the body of the ischium and its branch. The obturator artery, vein and nerve pass here, and the internal obturatorius muscle originates at the rim [29]. Close to the *acetabulum*, the cranial branch presents a cranially directed eminence (*eminentia iliopubica*) where the pectineus muscle originates (Figure 24C) [29]. The sharp cranial border of the cranial branch is the *pecten pubis* (Figure 24B). The rectus abdominis muscle inserts here [29].

The left and right hip bone are connected by the pelvic symphysis. The cranial half of this symphysis is formed by the united pubic bones and is therefore labelled as *symphysis pubica*. The caudal half is the *symphysis ischiadica* since both ischial bones join here (Figure 24B). More precisely, the symphyseal surfaces of the left and right caudal branches of the pubis (singular: *ramus caudalis ossis pubis facies symphysialis*), and the symphyseal surfaces of the left and right ischial branches (singular: *ramus ossis ischii facies symphysialis*) unite by means of cartilage (*synchondrosis*) that can ossify (*synostosis*). This can be noticed in Figure 24C. The pelvic symphysis demonstrates a ventrally projecting ridge (*crista symphysialis*) (Figure 24A). The cranial end of the pelvic symphysis presents a cranioventral tubercle. This is the *tuberculum pubicum ventrale* (Figure 24A,C). At the caudal side of the symphysis, the ischial arch (*arcus ischiadicus*) is the cranially directed arch in between the two ischial tuberosities (Figure 24B). 

The bony pelvis is formed when the united hip bones are joined with the sacrum. The pelvic cavity (*cavum pelvis*) is rather small with a narrow transverse diameter (*diameter transversa*) (Figure 24B). It has a cranial entrance and a caudal exit, i.e., *apertura pelvis cranialis* and *apertura pelvis caudalis*, respectively (Figure 24A). The width of the bony pelvis, measured between both cranial ventral iliac spines, was 12.6 cm in male skeleton 1, 13.3 cm in male skeleton 2, and 11.3 cm in the female skeleton. The length, measured from the iliac crest to the ischial tuberosity, was 17.2 cm in male skeleton 1, 17.9 cm in male skeleton 2, and 16.5 cm in the female skeleton.

#### 3.8.2. Femoral Skeleton (*Skeleton femoris*)

The femoral skeleton consists of the *os femoris* (femur), the *ossa sesamoidea musculi gastrocnemii*, and the *patella*. These will be discussed below in separate paragraphs.

##### Femur

The femur is a long, cylindrical bone that presents a curvature in cranial direction (Figure 25A). From the most proximal to the most distal extremity, the femur measured 23.2 cm in male skeleton 1, and 24.6 cm in male skeleton 2. The value for the female was 19.4 cm. Since a transection of the femoral body (*corpus femoris*) is round, no surfaces and margins can be identified. The proximal and distal ends of the femur are very characteristic. The proximal aspect of the femur is represented by the spherical head (*caput ossis femoris*) and the trochanters. The head is isolated from the body by a short neck (*collum ossis femoris*) (Figure 25A–D). It fits within the *acetabulum* and is attached to the acetabular fossa by a ligament that inserts into the central fovea of the femoral head (*fovea capitis*) (Figure 25C). The trochanters consist of the greater trochanter (*trochanter major*) and the lesser trochanter (*trochanter minor*). The greater trochanter is positioned at the lateral side of the head and extends proximal to it (Figure 25A–D). The lesser trochanter can be found caudodistal to the head (Figure 25B–D). Lateral to the head is a deep excavation in the base of the greater trochanter called the trochanteric fossa (*fossa trochanterica*) (Figure 25C,D). Both trochanters are connected by the intertrochanteric crest (*crista intertrochanterica*) (Figure 25D).

The irregular surface that can be seen on the femoral body caudolateral to the lesser trochanter is the gluteal tuberosity (*tuberositas glutea*). In species in which it is more prominent, this structure is called the third trochanter (*trochanter tertius*) [41]. The distal elongation of the lesser trochanter that forms a diagonal bony ridge on the femoral body serves as insertion site for the pectineus muscle [29], hence its name *linea pectinea*. The *linea aspera* is a rough line in the middle third of the shaft where several muscles such as the adductors insert (Figure 25D) [29].

The distal extremity of the femur is mainly shaped by the condyles and the *trochlea*. The medial condyle is larger than the lateral condyle and projects more distally (Figure 25B,C). Its lateral aspect possesses the medial epicondyle. Proximal to this epicondyle lies the medial supracondylar line (*linea supracondylaris medialis*). A small, smooth articular surface for the medial sesamoid bone of the gastrocnemius muscle can be observed at the caudal side of the femur, just proximal to the medial condyle (Figure 25C,D). In analogy with this *facies articularis sesamoidea medialis*, the slightly larger articular surface for the lateral sesamoid bone (*facies articularis sesamoidea lateralis*) can be recognized proximal to the lateral condyle. This condyle also presents an epicondyle, and a supracondylar line that runs proximal to the epicondyle (*linea supracondylaris lateralis*) (Figure 25A,D). Unique to the lateral condyle is the presence of a deep excavation on the lateral surface, directly caudal to the lateral epicondyle, where the popliteus muscle attaches [29]. This is the popliteal fossa (*fossa poplitea*) (Figure 25A). Another fossa is the intercondylar fossa (*fossa intercondylaris*), which sits in between both condyles and can be observed on a caudal view (Figure 25D). This view also allows for noting the popliteal surface (*facies poplitea*) (Figure 25D). On a cranial view, the trochlea (*trochlea ossis femoris*) can be studied. This grooved structure harbors a smooth surface for articulation with the patella (*facies articularis patellaris*). It is faintly asymmetrical as the lateral lip of the groove is marginally sharper and narrower than the medial lip (Figure 25B).

##### Sesamoid bones

Three sesamoid bones are associated with the femur. The larger of the three is the kneecap (*patella*) (Figure 26A,B). Moreover, it is the larger sesamoid bone of the entire skeleton. In general, the shape is rectangular with rounded corners. This may not be that clear when the cranial side (*facies cranialis*) is viewed but becomes more apparent when observing the articular surface (*facies articularis*) that slides within the femoral *trochlea*. However, the presence of the pyramidal *apex patellae*, which points in a distal direction, renders the *patella* an ovoid to triangular silhouette. This apex is connected to the tibial tuberosity by means of the patellar ligament. The proximal *basis patellae*, where the tendon of the femoral quadriceps muscle inserts, is slightly pointed. The lateral and medial margins run parallel to each other. The lateral and medial patellar ligaments attach here [29].

At the caudal side of the femur, just proximal to the condyles, sits a pair of small sesamoid bones. The lateral is the larger of the two. Not only has it a larger diameter, it is also higher in craniocaudal direction than the medial (Figure 26C). They are located within the tendons of the two heads of the gastrocnemius muscle. Thus, the left sesamoid bone (*os sesamoideum musculi gastrocnemii laterale*) and the right sesamoid bone (*os sesamoideum musculi gastrocnemii mediale*) are found in the proximal tendons of the lateral and medial head of the gastrocnemius muscle, respectively. However, they are not entirely embedded within these tendons. With their articular surfaces, the left and right sesamoid bone have contact with the respective articular surfaces on the femur (*facies articularis sesamoidea lateralis* and *facies articularis sesamoidea medialis*, respectively).

#### 3.8.3. *Skeleton cruris*

The crural skeleton (*skeleton cruris*) is composed of the tibia (*tibia*) and the fibula (*fibula*). The tibia is positioned at the medial side of the crural skeleton. It is the more prominent of the two bones. The fibula, which is located at the lateral side of the tibia, is much slimmer. The fibula proximally articulates with the tibia, but not with the femur. Only the tibia articulates with the femur (Figure 1B).

##### Tibia

The tibia measured 20.1 cm and 20.8 cm in length in male skeleton 1 and 2, respectively. The value for the female was 16.6 cm. The proximal end of the tibia articulates with the femur to form the femorotibial joint. More specifically, either of the femoral condyles articulates with the ipsilateral condyle of the tibia (Figure 27A–C). The area in between both condyles is the intercondylar area (*area intercondylaris*). It could be partitioned into the cranial, central, and caudal intercondylar area (*area intercondylaris cranialis*, *area intercondylaris centralis*, and *area intercondylaris caudalis*, respectively) (Figure 27B,D). The lateral and medial condyle each possess a proximal articular surface (*facies articularis proximalis*) that is slightly concave (Figure 27A–D). The concavity is emphasized by the presence of a tubercle at the medial rim of the articular surface. Since these tubercles are positioned in between both condyles, in the central intercondylar area, they can be nominated as the lateral and medial intercondylar tubercle (*tuberculum intercondylare laterale* and *tuberculum intercondylare mediale*, respectively). Both the lateral and medial intercondylar tubercles form the intercondylar eminence (*eminentia intercondylaris*) (Figure 27D). The concavity of the lateral articular surface is, however, less pronounced than that of the medial articular surface. Moreover, the surface is convex in a craniocaudal direction (Figure 27D). The fibular articular surface (*facies articularis fibularis*) sits at the lateral side of the lateral condyle. It forms the proximal tibiofibular joint with the head of the fibula (Figure 27A). The tuberosity at the craniodistal aspect of the cranial intercondylar area is the tibial tuberosity (*tuberositas tibiae*). The patellar ligament (*ligamentum patellae*) attaches here. The distal elongation of this structure is the tibial crest (*crista tibiae*) (Figure 27B,C). At the caudal side of the tibia, a groove is located in between both condyles. This is the popliteal notch (*incisura poplitea*) (Figure 27D).

The proximal half of the tibial body (*corpus tibiae*) presents a triangular cross section. This is largely due to the prominent tibial crest and the excavated extensor groove (*sulcus extensorius*) at the lateral side (Figure 27A). The distal half of the shaft is more tubular. A lateral, medial, and caudal surface can be identified on the tibial body (*corpus tibiae facies lateralis*, *medialis*, and *caudalis*, respectively). Proximally, the caudal surface presents a diagonal bony ridge for attachment of the popliteus muscle, i.e., the *linea musculi poplitei* (Figure 27D). The caudal surface is flanked by the lateral margin (*margo lateralis*) and the medial margin (*margo medialis*). The lateral margin faces the fibula and can therefore also be named the interosseous margin. The cranial margin (*margo cranialis*) is rather sharp as it lies distal to the tibial crest with the extensor groove at its lateral side (Figure 27A–D).

The distal end is characterized by the very prominent medial malleolus (*malleolus medialis*). It is elongated and curved to some extent (Figure 27B,D). A deep groove (*sulcus malleolaris*) can be observed at its caudal aspect (Figure 27D). A smooth articular surface lies at the lateral side of the medial malleolus (Figure 27B). It articulates with the articular surface for the medial malleolus on the talus. A much larger articular surface for articulation with the *trochlea* of the talus is present at the distal extremity of the tibia. This is the *cochlea tibiae*. The fibula does not form a distal tibiofibular joint, but a fibular notch (*incisura fibularis*) is present at the lateral side of the *cochlea*, where the fibula lies in close contact with the tibia (Figure 27A).

##### Fibula

The fibula is a long, slender bone with a proximal, relatively wide, lateromedially compressed head (*caput fibulae*), a narrower neck (*collum fibulae*), a rounder body (*corpus fibulae*) and a distal end that presents the pyramidal lateral malleolus (*malleolus lateralis*) (Figure 28A). From the head to the lateral malleolus, the fibula’s length was 17.9 cm in male skeleton 1, and 20.8 cm in male skeleton 2. The value for the female was 15.7 cm. The head articulates with the lateral condyle of the tibia by means of its medially oriented articular surface (*facies articularis capitis fibulae*) (Figure 28B and C). The body has a lateral surface (*facies lateralis*), a medial surface (*facies medialis*), and a caudal surface (*facies caudalis*) (Figure 28A,C,D). The lateral and medial surfaces are flanked by the cranial and caudal margins (*margo cranialis* and *margo caudalis*, respectively) (Figure 28A,C). The medial surface is directed towards the tibia and can therefore be termed *margo interosseus* (Figure 28C). The caudal surface is bordered by the lateral and medial margins (*margo lateralis* and *margo medialis*, respectively) (Figure 28D). The medially oriented malleolar articular surface (*facies articularis malleoli*) allows for the fibular distal extremity to articulate with lateral malleolar articular surface of the talus (Figure 28C).

#### 3.8.4. *Skeleton pedis*

The foot skeleton (*skeleton pedis*) consists of the *ossa tarsi*, which form the *tarsus*, the *ossa metatarsalia I-V*, and the *ossa digitorum pedis* with their associated sesamoid bones. The composition of the *tarsus* is very different from that of the carpus. The configuration of the rest of the foot skeleton, which is composed of the metatarsals and the digits, is very similar to that of the hand skeleton. From the calcaneus to the tip of the third digit, the length of the foot skeleton was 16.9 cm in male skeleton 1, and 17.7 cm in male skeleton 2. The value for the female was 13.1 cm.

There are seven tarsal bones that are arranged in three rows (Figure 29). The crural row consists of the medially located *talus* and the laterally positioned *calcaneus*. The intertarsal row is formed by the navicular bone (*os naviculare* or *os tarsi centrale*) at the medial side and the cuboid (*os cuboideum* or *os tarsale quartum*) at the opposite side. The *talus* rests on the proximal articular surface of the navicular bone. The *calcaneus* is supported by the cuboid bone. The metatarsal row is, from medial to lateral, occupied by the medial cuneiform bone (*os cuneiforme mediale* or *os tarsale primum*), the intermediate cuneiform bone (*os cuneiforme intermedium* or *os tarsale secundum*), the lateral cuneiform bone (*os cuneiforme laterale* or *os tarsale tertium*), and the cuboid. Indeed, this high tarsal bone is seated in both the intertarsal and the tarsal row. The medial cuneiform bone is relatively long. It forms the basis of the *hallux* as it articulates with the first metatarsal bone. The intermediate cuneiform bone is the smaller tarsal bone. It articulates with the second metatarsal bone. The third metatarsal bone, which is somewhat larger than its medial neighbor, shows articulation with the third metatarsal. The cuboid bone supports both the fourth and fifth metatarsal bones. Since the *talus* and *calcaneus* are the larger bones of the *tarsus*, they are described in detail.

##### Talus

The talus is shown in Figure 30. It can be partitioned into the large, proximal body (*corpus tali*), the small, middle neck (*collum tali*), and the distal head (*caput tali*). The body presents the talar trochlea (*trochlea tali*) for articulation with the tibial *cochlea*. The medial malleolus of this bone articulates with a specific crescent shaped articular surface for the medial malleolus (*facies articularis malleolaris medialis*), which is located at the medial side of the body. At the opposite side, the lateral malleolus, which is formed by the fibula articulates with the articular surface for the lateral malleolus (*facies articularis malleolaris lateralis*). The medial tubercle (*tuberculum mediale*) can be seen at the medial side of the trochlea, proximal to the articular surface for the medial malleolus. The head of the talus shows an articular surface for the navicular bone (*facies articularis navicularis*). These structures can be identified in Figure 30A.

The talus presents two articular surfaces for the calcaneus (*facies articulares calcaneae*), which sits at the plantar side of the talus. The proximal is larger and concave. The distal is smaller and rather flat. The calcanean groove (*sulcus calcanei*) is located in between both articular surfaces. The lateral tubercle (*tuberculum laterale*) can be observed at the lateral side of the proximal aspect of the *trochlea*. Plantar to the latter structure and medial to the proximal articular surface for the calcaneus is a groove for a flexor muscle of the hallux (*sulcus tendinis*) (Figure 30B). 

##### Calcaneus

The position of the calcaneus is lateral to the talus, since the latter tarsal bone rests on the navicular while the former tarsal bone articulates with the cuboid by means of a specific articular surface (*facies articularis cuboidea*) (Figure 29). However, the calcaneal tuber (*tuber calcanei*) is located plantar to the talus. The plantaris, gastrocnemius, and soleus muscles attach here. At the medial and lateral sides of the calcaneal tuber remain the medial process (*processus medialis*) and the lateral process (*processus lateralis*), respectively (Figure 31A,B). The proximal, middle, and distal articular surfaces for the talus (*facies articulares talares*) are at the dorsal side of the calcaneus (Figure 31A). The middle talal articular surface is placed on the *sustentaculum tali*. This structure, to which several ligaments attach, projects medially and sustains the talus (Figure 31A,B). A groove for the tendon of a flexor muscle for the hallux is visible at the plantar side of the *sustentaculum tali* (Figure 31B). In between the proximal and middle talal articular surfaces lies the calcaneal groove (*sulcus calcanei*) (Figure 31A). Finally, the plantar tubercle (*tuberculum plantare*) can be found at the lateral side, thus opposite the *sustentaculum tali* (Figure 31B).

The five metatarsal bones are cylindrical with an enlarged proximal base and distal head. A transection of the body is round. The base presents an articular surface for articulation with the tarsal bones. The head has a *trochlea* that fits within the articular fovea of the proximal phalanx (Figure 29A,B). At the plantar side of each *trochlea* sits a pair of oval sesamoid bones (Figure 29C).

In analogy with the skeleton of the hand, each of the digits of the foot has three phalanges, i.e., a proximal, a middle, and a distal, except for the *hallux* that possesses only two *phalanges hallucis*, i.e., the proximal phalanx and the distal phalanx. The middle *phalanx hallucis* is absent. The proximal and middle phalanges of digits II tot V are very similar in structure. The body of the middle phalanx is clearly shorter than that of the proximal phalanx. The base presents an articular *fovea* and the head shows a *trochlea* for articulation with the distal phalanx. The proximal articular *fovea* of the distal phalanx corresponds with the *trochlea* of the previous phalanx. The distal phalanges of the second to fifth digit are pointed and have a minute tuberosity (Figure 29A,B). The distal phalanx of the *hallux* is undoubtedly broader, and the tuberosity is much more pronounced (Figure 29A).

When the total lengths of the digits are compared, the following digital formula can be deduced: III > IV > II > V > I. The *hallux* is the shorter of the five digits, for the same reasons that have been mentioned in the description of the skeleton of the hand.

## 4. Discussion

According to Sayers and coworkers [42], the African-origin baboon is an underutilized nonhuman primate model that can recapitulate the conditions of human lifespan at the intersection of infectious diseases, inflammation, and human diseases of the brain, central organs, and metabolism. Indisputably, this statement promotes the increased use of the baboon in the laboratory environment. However, a sound knowledge of the anatomy and physiology of the baboon is a prerequisite for the translation of experimental data to man. Although the atlas by Swindler and Wood [29] greatly expands our knowledge of the comparative anatomy between the baboon and man, the detail present in the osteology chapter is disappointing. Due to the nature of this work—an atlas of primate gross anatomy—it is unfeasible to fully elaborate each anatomical system of the covered primate species, i.e., the baboon, chimpanzee, and man. The authors have deliberately chosen to discuss the fundamentals and analyze the dissimilarities between the species.

Our in-depth work on the osteology of the mantled baboon was initiated after researchers working at the BPRC were in demand of detailed descriptions of the anatomy of Old World monkeys. Not only would the meticulous depictions be valuable for the investigators, the veterinarians who are responsible for the medical care and welfare of the captive primates would also benefit from numerous detailed anatomical photographs [43]. When treating wounds or performing surgery, the veterinary professionals primarily rely on their knowledge of domestic mammal anatomy. Therefore, we have described the baboon as a mammal stricto sensu and not a human. This also makes sense from a locomotor point of view. The baboon is a quadrupedal animal, like a horse or a dog, while humans are bipedal [29]. When describing the anatomy of the human arm, the arm is studied in a supinated position, the palm remaining visible on a frontal view with the thumb pointing in lateral direction [44,45]. In contrast, quadrupedal animals are described in a walking position. This means that the thumb sits at the medial side, the anterior side of man is the ventral side of the animal, the term posterior translates as dorsal, superior as cranial, and inferior as caudal. Structures on the head are located rostral to more caudal structures when they are nearer to the tip of the nose. 

Since humans and non-human primates, such as the baboon, share a common ancestor, remarkable anatomical and physiological similarities are shared. These similarities have allowed and will allow for the baboon to play an important role as a model for humans [42]. However, several significant discrepancies were observed when studying the baboon skeletons and comparing these with a human skeleton present in the Morphology Museum of Ghent University. In addition, images in human anatomy atlases were consulted. Thus, the anatomy of the baboon does not appear to be identical to human anatomy. Because this conclusion can also be drawn when consulting the atlas by Swindler and Wood [29], below, we only present the major differences between the skeletons of baboons and humans. Some anatomical traits of the baboon skeleton that are not found in the skeletons of domestic mammals are additionally reviewed. In this way, our findings are put in perspective. They can point the pros and cons of utilizing the baboon in biomedical research that involves the skeleton.

The baboon skull presents a smaller neurocranium and, particularly, the frontal bone is smaller compared to man and not vaulted. Nevertheless, the supraorbital ridges are pronounced. These are reduced in humans, and especially in women [44]. The relatively larger orbits of the baboon have, as in humans, margins that are fully ossified, in contrast to domestic carnivores that present the lateral orbital ligament [46]. The *splanchnocranium* is elongated. Thus, the nasal bones and maxillae are long, forming the snout, which is comparable to domestic mammals. This short enumeration indicates that the general configuration of the skull differs unambiguously between baboons and humans. Yet, only a small number of significantly altered structures can be identified. For example, the reduced cribrous plate can only be observed through the small olfactory foramen in the baboon. Another example can be found in the orbital fissures, which are less extensive in the baboon. The dental formula of the baboon is identical to that of man. The canines are, however, larger in the baboon, and especially in the males they can be tusk-like (Figure 1).

The vertebrae of the baboon include seven cervical vertebrae, a number that is as good as constant in mammals, twelve or thirteen thoracic vertebrae, six or seven lumbar vertebrae, three fused sacral vertebrae, and twenty-ish caudal vertebrae. The sum of the thoracic and lumbar vertebrae is nineteen [29]. Thus, when twelve thoracic vertebrae are present, seven lumbar vertebrae can be identified. When thirteen thoracic vertebrae are counted, there are six lumbar vertebrae. The last thoracic, or first lumbar vertebra can be a transitional vertebra. This was seen in one of the examined skeletons. The first lumbar vertebra unilaterally presented a short floating rib instead of a short transverse process. In humans, twelve thoracic vertebrae, five lumbar vertebrae, five fused sacral vertebrae and four caudal vertebrae, which are fused to form the *os coccyx*, are present [44,45]. Most domestic mammals have thirteen thoracic vertebrae, except the horse, which has eighteen. In smaller domestic mammals, there are seven lumbar vertebrae and six in larger domestic mammals. The number of fused sacral vertebrae is three and five, respectively.

Similar to domestic mammals, the sternum of the baboon is composed of individual sternebrae that are joined by cartilage to form one unity. As a result, the manubrium and xyphoid are easily identified. In man, the sternebrae, and also the manubrium and xyphoid, are fused by means of bony tissue. No synchondroses but synostoses are present. 

The thoracic limb of the baboon resembles that of humans to a high extent. The same bones can be recognized. In domestic mammals, the clavicle can only be observed in a reduced form in the cat and the rabbit. It is fully formed in rodents. These have the presence of the central carpal bone in common with the baboon [35]. In humans, this bone has been fused with the scaphoid during fetal life [29]. 

Also, the pelvic limb is analogous in the baboon and man. However, the hip of the baboon is much longer and slenderer, and the transverse diameter narrower in comparison with man. The human hip has undergone modifications to accommodate with the bipedal stride. In this view, the hip of the baboon resembles better the hip of the domestic mammals.

A penile bone is present in the male hamadryas baboon. In the domestic mammals, this bone is very considerable in Canidae, petite in Felidae, and absent in the pig, the horse, and the ruminants. Compared to the penile bone of the dog, the penile bone of the hamadryas baboon is diminutive [41]. The presence of the penile bone is a typical trait of primates. We have previously described the penile bone in another commonly used primate in biomedical research, i.e., the common marmoset (*Callithrix jacchus*) [35,38]. It is very interesting to note that although humans are primates, men lack the penile bone. It seems that the penile bone was lost during evolution. The reasons remain, however, unclear [47].

A shortcoming in the present study is the inclusion of only two male baboon skeletons and one female baboon skeleton. We have therefore limited the comparison between the skeletons of both sexes to the dimensions of the skull and mandible, the various segments of the vertebral column, the sternum, the clavicle, the bony pelvis, the long bones of the appendicular skeleton, and the hand and foot skeleton. It can be concluded from the data that the female skeleton is smaller than the male skeletons. It was already mentioned in the introduction that the male’s body measures approximately 65 cm in length and that females are only half the size. This statement is only partially reflected by our measurements. The female was indeed smaller than the males, but the difference was not that large. However, our measurements have been performed on skeletons, and especially the lengths of the various vertebral segments can be influenced by the technique of reassembling the skeletons.

Future studies should include the investigation of more baboon skeletons, of both sexes, and focus on the potential morphological and morphometric differences between male and female baboon skeletons. In humans, a number of distinctions between the skeletons of women and men have been described, and these allow for the differentiation between both sexes [48]. For example, it is known that the pelvic cavity is rounder in women compared to men [48]. In the baboon, the skull, especially, has already been examined in the potential differentiation of male vs. female skeletons [49]. In addition, the lengths and weights of the metacarpals and metatarsals have been investigated in the mantled baboon since the relative sizes of the bones of the hand and foot may provide helpful supplemental information on the relations between primate species, including humans [50]. It could therefore be worthwhile to set up a detailed, comparative study in which morphological and morphometric data are collected in view of the potential differentiation between male and female baboon skeletons. The value of such data could perhaps also be found in the field of anthropology as the species is important in the evolution of neogene cercopithecids [21,51].

Given the widespread use of radiography and computed tomography scanning (CT-scanning) in veterinary medicine, another future research project could include these medical imaging techniques to characterize the baboon skeleton [52,53,54]. The present manuscript has provided the morphological basis to take the step to medical imaging. It will allow for the correct interpretation of radiographs and CT-scans.

## 5. Conclusions

This article provides osteological data on the mantled baboon in order to facilitate the translation of experimental data obtained in this species to human medicine, and to assist veterinarians during, for example, wound treatment or surgical interventions. The aim of the present investigation was to describe the baboon skeleton in great detail, complementing existing works, and not to perform a comparative study on the skeletons of the baboon and man. For this matter, we kindly refer to the work of Swindler and Wood [29]. However, we have discussed the major anatomical differences between the two species. Since these are not numerous and far from substantial, the baboon appears to be a valuable model for humans. Future studies could focus on the joints and the musculature of the mantled baboon to obtain a complete anatomical atlas of the locomotor system. In addition, our osteological study could be revisited with the inclusion of more male and female baboon skeletons, focusing on the potential differences between both sexes.

## Figures and Tables

**Figure 1 animals-13-03124-f001:**
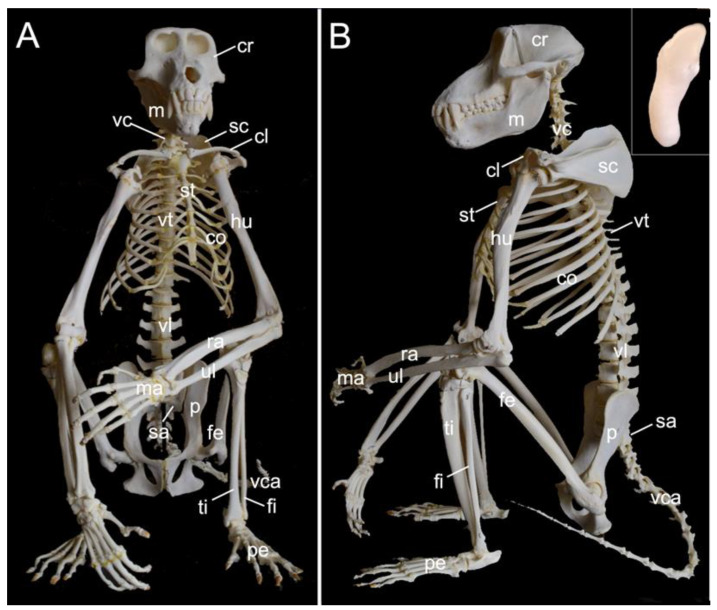
Skeleton of a male hamadryas baboon in seated position. (**A**) Ventral view, (**B**) Left lateral view. The costal cartilages of the last three pairs of ribs were not fully recovered during the mounting of the skeleton. Insert: Penile bone (*os penis*), lateral view. cl: *clavicula*, co: *costae*, cr: *cranium*, fe: *femur*, fi: *fibula*, hu: *humerus*, m: *mandibula*, ma: *manus*, p: *pelvis*, pe: *pes*, ra: *radius*, sa: *sacrum*, sc: *scapula*, st: *sternum*, ti: *tibia*, ul: *ulna*, vc: *vertebrae cervicales*, vca: *vertebrae caudales*, vl: *vertebrae lumbales*, vt: *vertebrae thoracicae*.

**Figure 2 animals-13-03124-f002:**
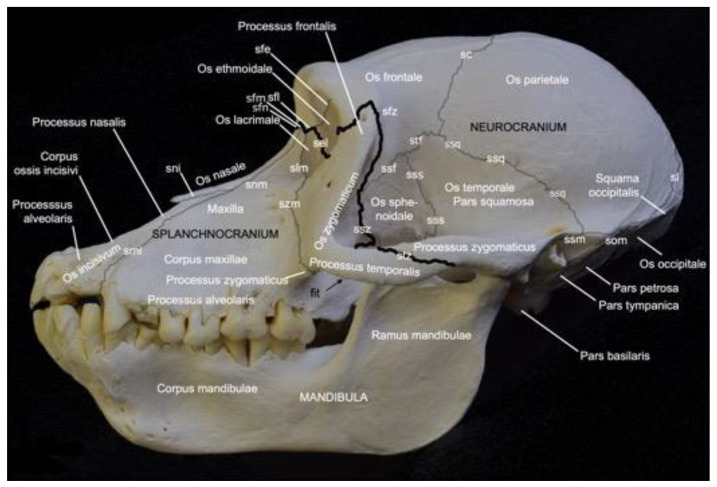
*Ossa cranii et faciei* of the skull of the female hamadryas baboon, left lateral view. The sutures between the cranial bones are highlighted in grey. The sutures between the *splanchnocranium* and the *neurocranium* are highlighted in black. Where appropriate, the processes and specific parts of the bones are indicated. Notice the irregularly positioned and worn off incisors and canines. fit: *fossa infratemporalis*, sc: *sutura coronalis*, sel: *sutura ethmoidolacrimalis*, sfe: *sutura frontoethmoidalis*, sfl: *sutura frontolacrimalis*, sfm: *sutura frontomaxillaris*, sfn: *sutura frontonasalis*, sfz: *sutura frontozygomatica*, sl: *sutura lambdoidea*, slm: *sutura lacrimomaxillaris*, smi: *sutura maxilloincisiva*, sni: *sutura nasoincisiva*, snm: *sutura nasomaxillaris*, som: *sutura occipitomastoidea*, ssf: *sutura sphenofrontalis*, ssm: *sutura squamomastoidea*, ssp: *sutura sphenoparietalis*, ssq: *sutura squamosa*, sss: *sutura sphenosquamosa*, ssz: *sutura sphenozygomatica*, stf: *sutura temporofrontalis*, stz: *sutura tympanozygomatica*, szm: *sutura zygomaticomaxillaris*.

**Figure 3 animals-13-03124-f003:**
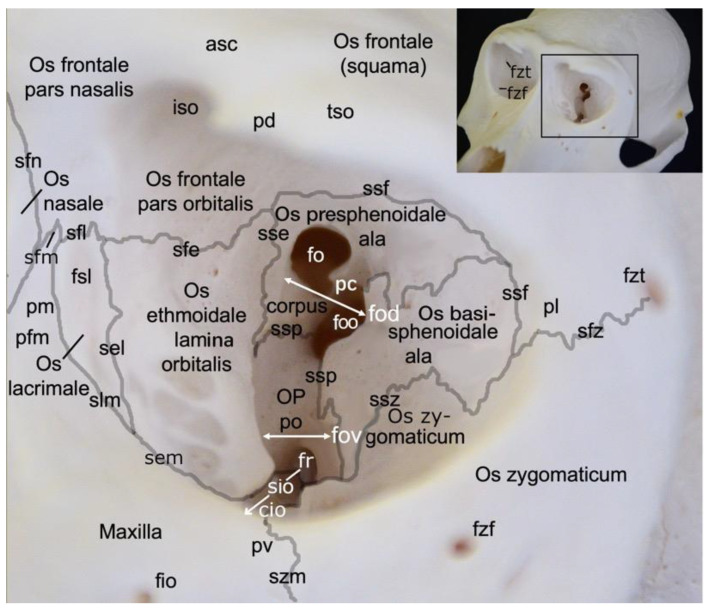
Left orbit of the hamadryas baboon. The exact location of this enlarged view is shown in the insert. asc: *arcus supraciliaris*, cio: *canalis infraorbitalis*, fio: *foramina infraorbitalia*, fo: *foramen opticum*, fod: *fissura orbitalis dorsalis*, foo: *foramen orbitale*, fov: *fissura orbitalis ventralis*, fr: *foramen rotundum*, fsl: *fossa sacci lacrimalis*, fzf: *foramen zygomaticofaciale*, fzt: *foramen zygomaticotemporale*, iso: *incisura supraorbitalis*, OP: *os palatinum*, pc: *processus clinoideus*, pd: *paries dorsalis*, pfm: *processus frontalis maxillae*, pl: *paries lateralis*, pm: *paries medialis*, po: *processus orbitalis*, pv: *paries ventralis*, sel: *sutura ethmoidolacrimalis*, sfe: *sutura frontoethmoidalis*, sfl: *sutura frontolacrimalis*, sfm: *sutura frontomaxillaris*, sfn: *sutura frontonasalis*, sfz: *sutura frontozygomatica*, sio: *sulcus infraorbitalis*, slm: *sutura lacrimomaxillaris*, sse: *sutura sphenoethmoidalis*, ssf: *sutura sphenofrontalis*, ssp: *sutura sphenopalatina*, ssz: *sutura sphenozygomatica*, szm: *sutura zygomaticomaxillaris*, tso: *torus supraorbitalis*.

**Figure 4 animals-13-03124-f004:**
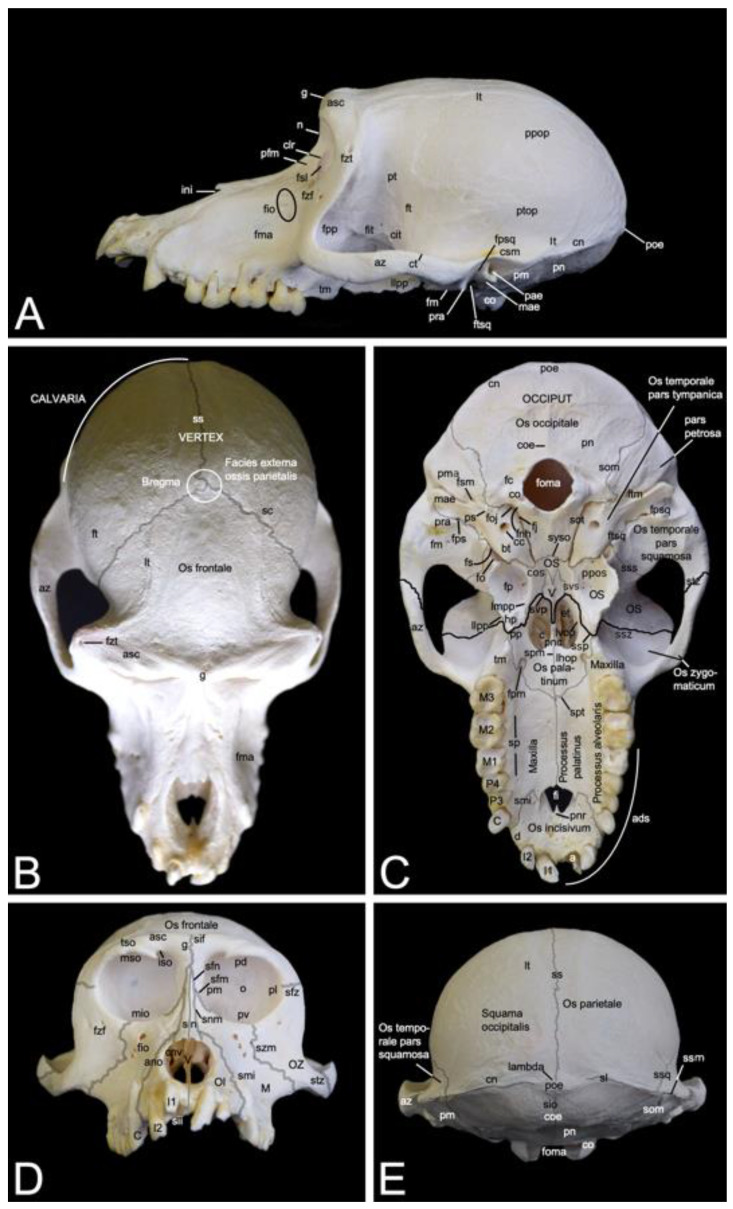
Skull of the female hamadryas baboon. (**A**) Left lateral view, (**B**) Dorsal view, (**C**) Ventral view, (**D**) Rostral view, (**E**) Caudal view. Notice the irregularly positioned and worn off incisors and canines. a: *alveolus*, ads: *arcus dentalis superior*, ano: *apertura nasi ossea*, asc: *arcus supraciliaris*, az: *arcus zygomaticus*, bt: *bulla tympanica*, c: *choana*, C: *dens caninus*, cc: *canalis caroticus*, cit: *crista infratemporalis*, clr: *crista lacrimalis rostralis*, cn: *crista nuchae*, cnv: *concha nasalis ventralis*, co*: condylus occipitalis*, coe: *crista occipitalis externa*, cos: *corpus ossis sphenoidalis*, csm: *crista supramastoidea*, ct: *crista temporalis*, d: *diastema*, et: *ethmoturbinalia* from *os ethmoidale*, fc: *fossa condylaris*, fi: *foramen incisivum*, fio: *foramina infraorbitalia*, fit: *fossa infratemporalis*, fj: *fossa jugularis*, fm: *fossa mandibularis*, fma: *fossa maxillaris*, fnh: *foramen nervi hypoglossi*, fo: *foramen ovale*, foj: *foramen jugulare*, foma: *foramen magnum*, fp: *fossa pterygoidea*, fpm: *foramen palatinum majus*, fpp: *fossa pterygopalatina*, fps: *foramen petrosquamosum*, fpsq: *fissura petrosquamosa*, fs: *foramen spinosum*, fsl: *fossa sacci lacrimalis*, fsm: *foramen stylomastoideum*, ft: *fossa temporalis*, ftm: *fissura tympanomastoidea*, ftsq: *fissura tympanosquamosa*, fzf: *foramen zygomaticofrontale*, fzt: *foramen zygomaticotemporale*, g: *glabella*, hp: *hamulus pterygoideus*, I1: *dens incisivus primus*, I2: *dens incisivus secundus*, ini: *incisura nasoincisiva*, iso: *incisura supraorbitalis*, lhop: *lamina horizontalis ossis palatini*, llpp: *lamina lateralis processus pterygoidei*, lmpp: *lamina medialis processus pterygoidei*, lt: *linea temporalis*, lvop: *lamina verticalis ossis palatini*, M: *maxilla*, M1: *dens molaris primus*, M2: *dens molaris secundus*, M3: *dens molaris tertius*, mae: *meatus acusticus externus*, mio: *margo infraorbitalis*, mso: *margo supraorbitalis*, n: *nasion*, o: *orbita*, OI: *os incisivum*, OS*: os sphenoidale*, OZ: *os zygomaticum*, P3: *dens premolaris tertius*, P4: *dens premolaris quartus*, pae: *porus acusticus externus*, pd: *paries dorsalis*, pfm: *processus frontalis maxillae*, pl: *paries lateralis*, pm: *paries medialis*, pma: *processus mastoideus*, pn: *planum nuchale*, pnc: *processus nasalis caudalis*, pnr: *processus nasalis rostralis*, poe: *protuberantia occipitalis externa* (*inion*), pp: *processus pyramidalis*, ppop: *planum parietale ossis parietalis*, ppos: *processus pterygoideus ossis sphenoidalis*, pra: *processus retroarticularis*, ps: *processus styloideus*, pt: *pterion*, ptop: *planum temporale ossis parietalis*, pv: *paries ventralis*, sc: *sutura coronalis*, sfm: *sutura frontomaxillaris*, sfn: *sutura frontonasalis*, sfz: *sutura frontozygomatica*, sif: *sutura interfrontalis*, sii: *sutura interincisiva*, sin: *sutura internasalis*, sio: *sutura interoccipitalis*, sl: *sutura lambdoidea*, smi: *sutura maxilloincisiva*, snm: *sutura nasomaxillaris*, som: *sutura occipitomastoidea*, sot: *sutura occipitotympanica*, sp: *sulcus palatinus*, spm: *sutura palatina mediana*, spt: *sutura palatina transversa*, ss: *sutura sagittalis*, ssm: *sutura squamomastoidea*, ssp: *sutura sphenopalatina*, ssq: *sutura squamosa*, sss: *sutura sphenosquamosa*, ssz: *sutura sphenozygomatica*, stz: *sutura tympanozygomatica*, svp: *sutura vomeropalatina*, svs: *sutura vomerosphenoidalis*, syso: *synchondrosis sphenooccipitalis*, szm: *sutura zygomaticomaxillaris*, tm: *tuber maxillae*, tso: *torus supraorbitalis*, V: *vomer*.

**Figure 5 animals-13-03124-f005:**
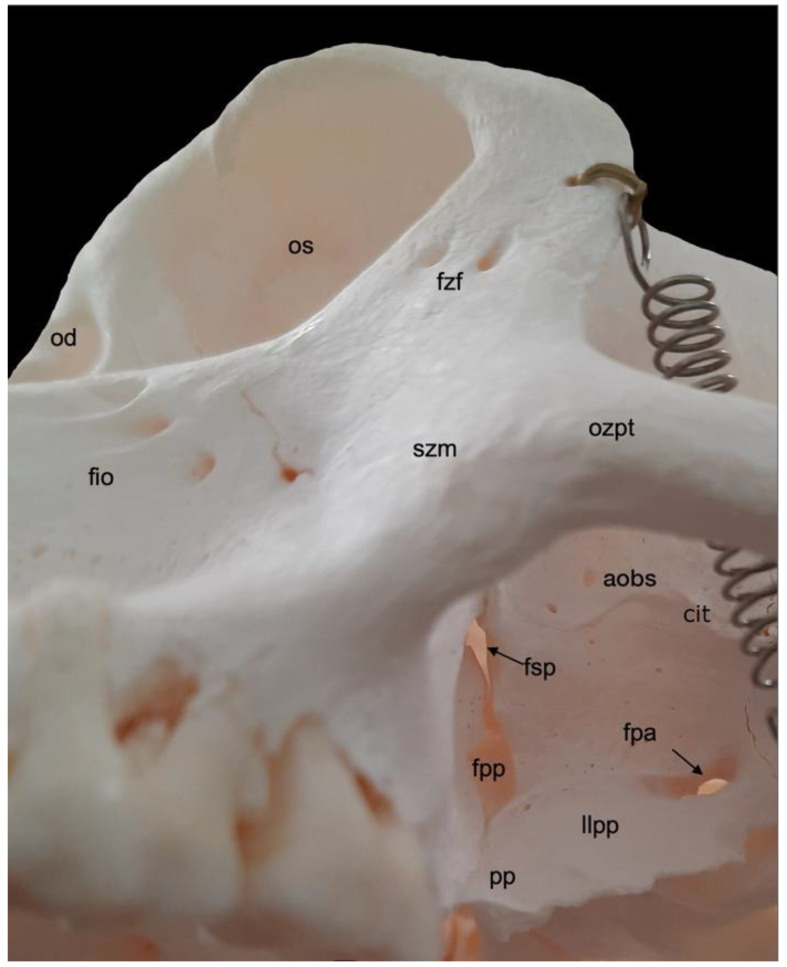
Left ventrolateral view of the infratemporal fossa of the hamadryas baboon. *aobs: ala ossis basisphenoidalis*, cit: *crista infratemporalis*, fio: *foramina infraorbitalia*, fpa: *foramen pterygoalare*, fpp: *fissura pterygopalatina*, fsp: *foramen sphenopalatinum*, fzf: *foramina zygomaticofrontalia*, llpp: *lamina lateralis processus pterygoidei*, od: *orbita dextra*, os: *orbita sinistra*, ozpt: *os zygomaticum processus temporalis*, pp: *processus pyramidalis*, szm: *sutura zygomaticomaxillaris*.

**Figure 6 animals-13-03124-f006:**
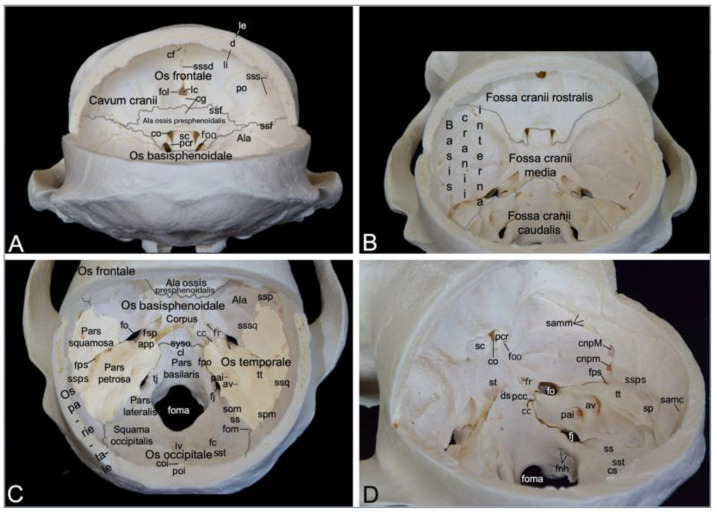
Cranial cavity of the hamadryas baboon. The calvaria has been removed. (**A**) Caudal view, (**B**) Caudodorsal view, (**C**) Dorsal view, (**D**) Left caudodorsal view. app: *apex partis petrosae*, av: *apertura externa aqueductus vestibuli*, cc: *canalis caroticus*, cf: *crista frontalis*, cg: *crista galli*, cl: *clivus*, cnpM: *canalis n. petrosi majoris*, cnpm: *canalis n. petrosi minoris*, co: *canalis opticus*, coi: *crista occipitalis interna*, cs: *confluens sinuum*, d: *diploë*, ds: *dorsum sellae*, fc: *fossa cerebellaris*, fj: *foramen jugulare*, fnh: *foramen n. hypoglossi*, fo: *foramen ovale*, fol: *foramen olfactorium*, fom: *foramen mastoideum*, foma: *foramen magnum*, foo: *foramen orbitale*, fpo: *fissura petrooccipitalis*, fps: *fissura petrosquamosa* with *foramen petrosquamosum*, fr: *foramen rotundum*, fsp: *fissura sphenopetrosa*, iv: *impressio vermialis*, lc: *lamina cribrosa*, le: *lamina externa*, li: *lamina interna*, pai: *porus acusticus internus*, pcc: *processus clinoideus caudalis*, pcr: *processus clinoideus rostralis*, po: *planum orbitale*, poi: *protuberantia occipitalis interna*, samc: *sulcus arteriae meningeae caudalis*, samm: *sulcus arteriae meningeae mediae*, sc: *sulcus chiasmatis*, som: *sutura occipitomastoidea*, sp: *sulcus sinus prootici*, spm: *sutura parietomastoidea*, ss: *sulcus sinus sigmoidei*, ssf: *sutura sphenofrontalis*, ssp: *sutura sphenoparietalis*, ssps: *sulcus sinus petrosquamosi*, ssq: *sutura squamosa*, sss: *sulcus sinus sphenoparietalis*, sssd: *sulcus sinus sagittalis dorsalis*, sssq: *sutura sphenosquamosa*, sst: *sulcus sinus transversi*, st: *sella turcica*, syso: *synchondrosis sphenooccipitalis*, tj: *tuberculum jugulare*, tt: *tegmentum tympani*.

**Figure 7 animals-13-03124-f007:**
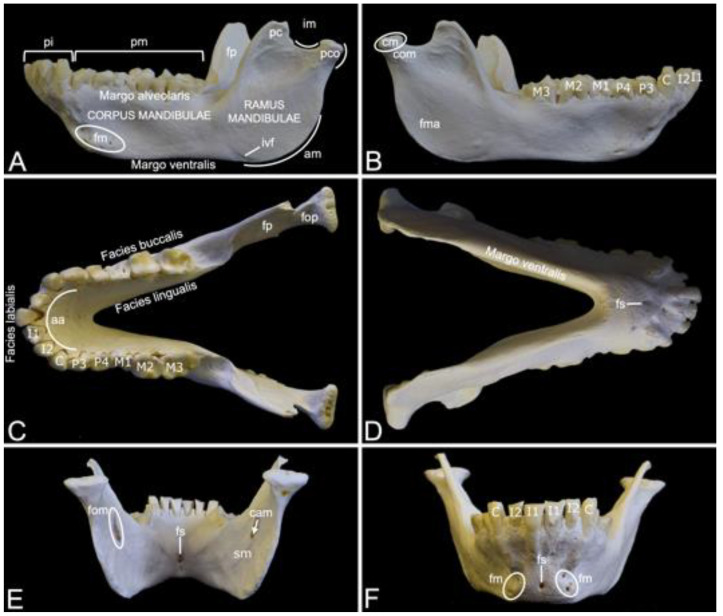
Mandible of the female hamadryas baboon. (**A**) Left lateral view, (**B**) Right lateral view, (**C**) Dorsal view, (**D**) Ventral view, (**E**) Caudal view, (**F**) Rostral view. aa: *arcus alveolaris*, am: *angulus mandibulae*, C: *dens caninus*, cm: *caput mandibulae*, cam: *canalis mandibulae*, com: *collum mandibulae*, fm: *foramina mentalia*, fma: *fossa masseterica*, fom: *foramen mandibulae*, fop: *fovea pterygoidea*, fp: *fossa pterygoidea*, fs: *foramen symphysialis*, I1: *dens incisivus primus*, I2: *dens incisivus secundus*, im: *incisura mandibulae*, ivf: *incisura vasorum facialium*, M1: *dens molaris primus*, M2: *dens molaris secundus*, M3: *dens molaris tertius*, P3: *dens premolaris tertius*, P4: *dens premolaris quartus*, pc: *processus coronoideus*, pco: *processus condylaris*, pi: *pars incisiva*, pm: *pars molaris*, sm: *sulcus mylohyoideus*.

**Figure 8 animals-13-03124-f008:**
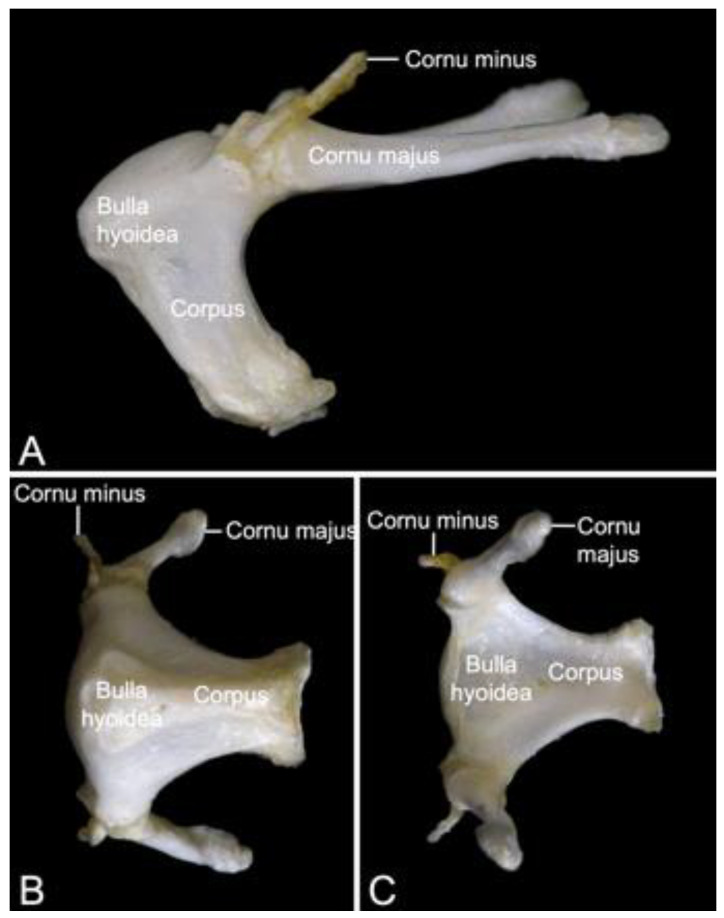
Hyoid bone of the hamadryas baboon. (**A**) Left lateral view, (**B**) Ventral view, (**C**) Dorsal view.

**Figure 9 animals-13-03124-f009:**
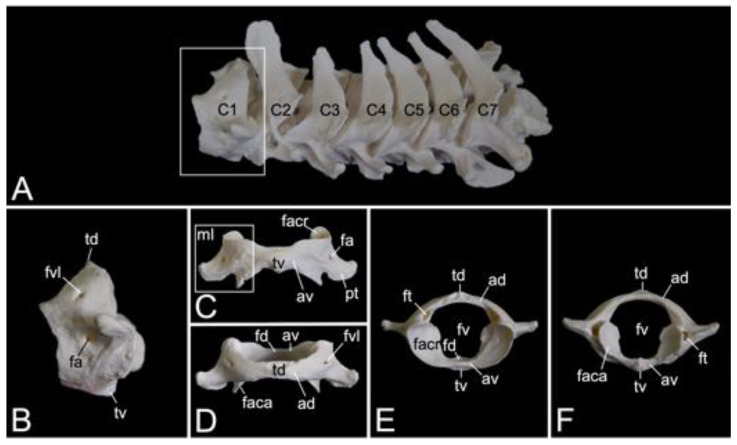
*Atlas* of the hamadryas baboon. (**A**) Left dorsolateral view of the cervical vertebral segment, (**B**) Left lateral view of the *atlas*, (**C**) Ventral view of the *atlas*, (**D**) Dorsal view of the *atlas*, (**E**) Cranial view of the *atlas*, (**F**) Caudal view of the *atlas*. ad: *arcus dorsalis*, av: *arcus ventralis*, fa: *foramen alara*, faca: *fovea articularis caudalis*, facr: *fovea articularis cranialis*, fd: *fovea dentis*, ft: *foramen transversarium*, fv: *foramen vertebrale*, fvl: *foramen vertebrale laterale*, ml: *massa lateralis*, pt: *processus transversus*, td: *tuberculum dorsale*, tv: *tuberculum ventrale*.

**Figure 10 animals-13-03124-f010:**
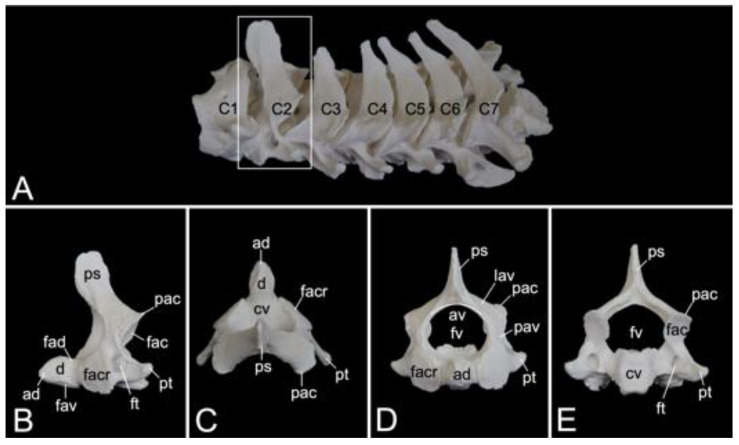
*Axis* of the hamadryas baboon. (**A**) Left dorsolateral view of the cervical vertebral segment, (**B**) Left lateral view of the *axis*, (**C**) Dorsal view of the *axis*, (**D**) Cranial view of the *axis*, (**E**) Caudal view of the *axis*. ad: *apex dentis*, av: *arcus vertebrae*, cv: *corpus vertebrae*, d: *dens (axis)*, fac: *facies articularis caudalis*, facr: *facies articularis cranialis*, fad: *facies articularis dorsalis*, fav: *facies articularis ventralis*, ft: *foramen transversarium*, fv: *foramen vertebrale*, lav*: lamina arcus vertebrae* pac: *processus articularis caudalis*, pav: *pediculus arcus vertebrae*, ps: *processus spinosus*, pt: *processus transversus*.

**Figure 11 animals-13-03124-f011:**
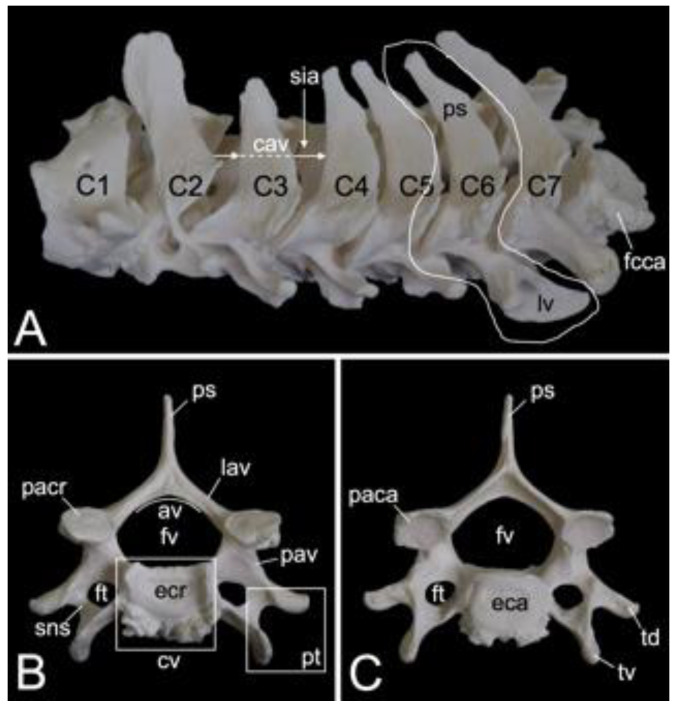
Sixth cervical vertebra of the hamadryas baboon. (**A**) Left dorsolateral view of the cervical vertebral segment, (**B**) Cranial view of the sixth cervical vertebra, (**C**) Caudal view of the sixth cervical vertebra. av: *arcus vertebrae*, C: *vertebra cervicalis*, cav: *canalis vertebralis*, cv: *corpus vertebrae*, eca: *extremitas caudalis*, ecr: *extremitas cranialis*, fcca: *fovea costalis caudalis*, ft: *foramen transversarium*, fv: *foramen vertebrale*, lav: *lamina arcus vertebrae*, lv: *lamina ventralis*, paca: *processus articularis caudalis*, pacr: *processus articularis cranialis*, pav: *pediculus arcus vertebrae*, ps: *processus spinosus*, pt: *processus transversus*, sia: *spatium interarcuale*, sns: *sulcus nervi spinalis*, td: *tuberculum dorsale*, tv: *tuberculum ventrale*.

**Figure 12 animals-13-03124-f012:**
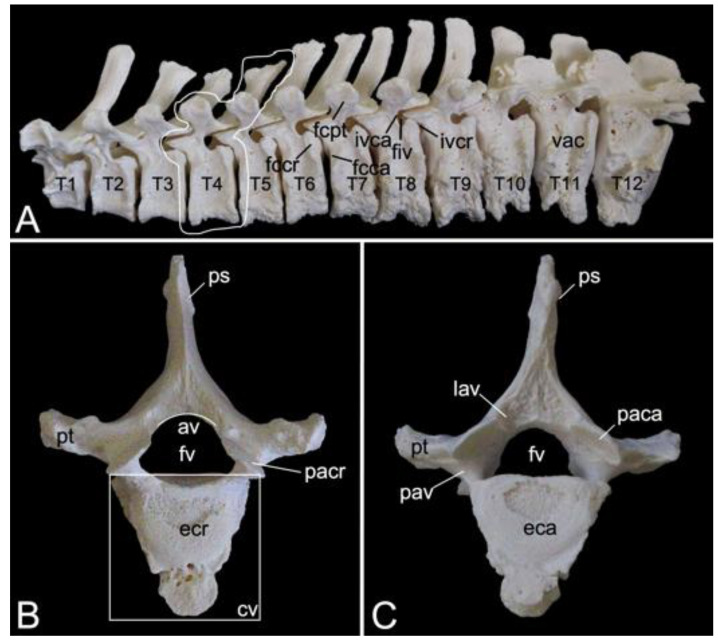
Thoracic vertebrae of the hamadryas baboon. (**A**) Left lateral view of the thoracic vertebral segment. (**B**) Cranial view of the fourth thoracic vertebra, (**C**) Caudal view of the fourth thoracic vertebra. Notice the arthrosis on vertebral bodies 5–12. av: *arcus vertebrae*, cv: *corpus vertebrae*, eca: *extremitas caudalis*, ecr: *extremitas cranialis*, fcca: *fovea costalis caudalis*, fccr: *fovea costalis cranialis*, fcpt: *fovea costalis processus transversi*, fiv: *foramen intervertebrale*, fv: *foramen vertebrale*, ivca: *incisura vertebralis caudalis*, ivcr: *incisura vertebralis cranialis*, lav: *lamina arcus vertebrae*, paca: *processus articularis caudalis*, pacr: *processus articularis cranialis*, pav: *pediculus arcus vertebrae*, ps: *processus spinosus*, pt: *processus transversus*, T: *vertebra thoracica*, vac: *vertebra anticlinalis*.

**Figure 13 animals-13-03124-f013:**
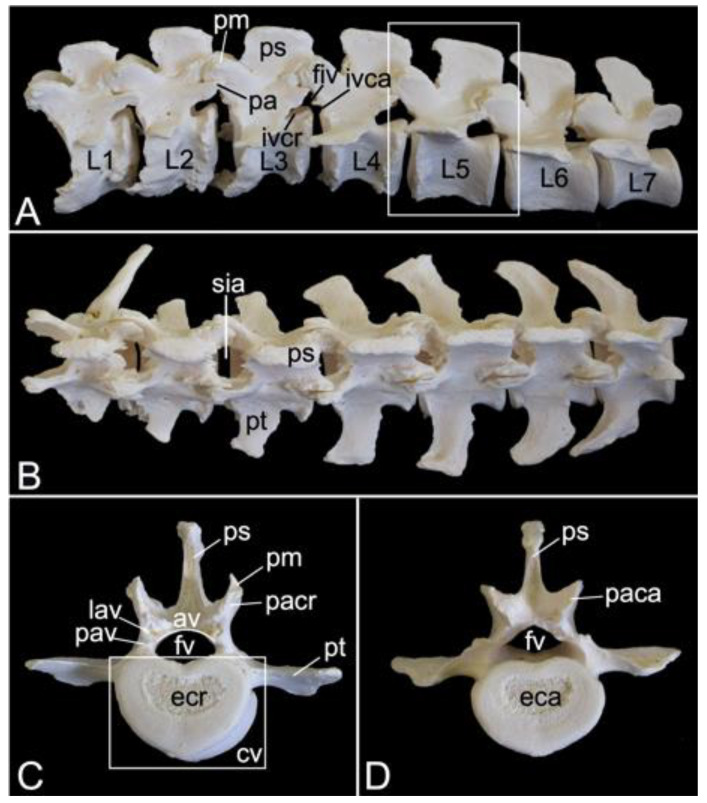
Lumbar vertebrae of the hamadryas baboon. (**A**) Left lateral view of the lumbar vertebral segment, (**B**) Dorsal view of the lumbar vertebral segment. Notice the transitional vertebra T13/L1, which is characterized by a pronounced right transverse process resembling a short rib, (**C**) Cranial view of the fourth lumbar vertebra, (**D**) Caudal view of the fourth lumbar vertebra. av: *arcus vertebrae*, cv: *corpus vertebrae*, eca: *extremitas caudalis*, ecr: *extremitas cranialis*, fv: *foramen vertebrale*, fiv: *foramen intervertebrale*, ivca: *incisura vertebralis caudalis*, ivcr: *incisura vertebralis cranialis*, L: *vertebra lumbalis*, lav: *lamina arcus vertebrae*, pa: *processus accessorius*, paca: *processus articularis caudalis*, pacr: *processus articularis cranialis*, pav: *pediculus arcus vertebrae*, pm: *processus mamillaris*, ps: *processus spinosus*, pt: *processus transversus*, sia: *spatium interarcuale*.

**Figure 14 animals-13-03124-f014:**
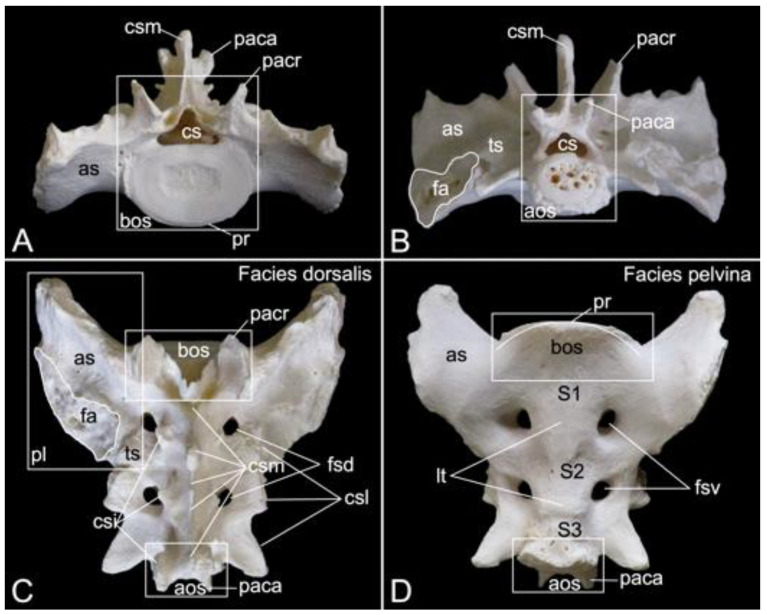
Sacrum of the hamadryas baboon. (**A**) Cranial view, (**B**) Caudal view, (**C**) Dorsal view, (**D**) Ventral view. aos: *apex ossis sacri*, as: *ala sacralis*, bos: *basis ossis sacri*, cs: *canalis sacralis*, csi: *crista sacralis intermedia*, csl: *crista sacralis lateralis*, csm: *crista sacralis mediana*, fa: *facies auricularis*, fsd: *foramina sacralia dorsalia*, fsv: *foramina sacralia ventralia*, lt: *lineae transversae*, paca: *processus articularis caudalis*, pacr: *processus articularis cranialis*, pl: *pars lateralis*, pr: *promontorium*, S: *vertebra sacralis*, ts: *tuberositas sacralis*.

**Figure 15 animals-13-03124-f015:**
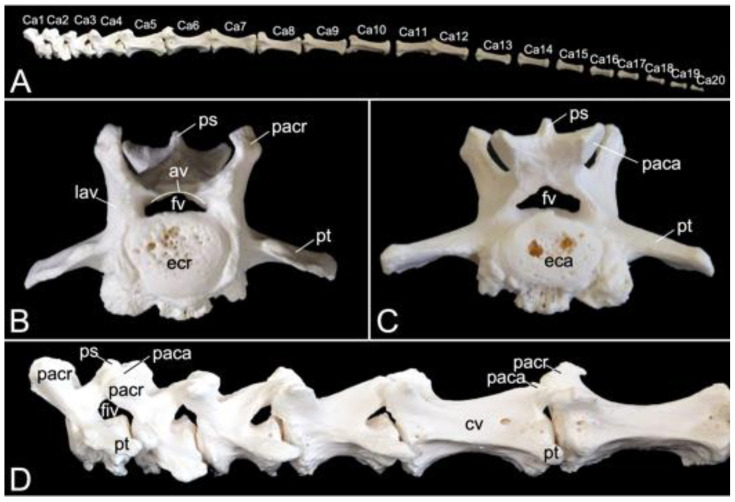
Osseous tail of the hamadryas baboon. (**A**) Lateral view, (**B**) Cranial view of the first caudal vertebra, (**C**) Caudal view of the first caudal vertebra, (**D**) Left lateral view of the first six caudal vertebrae. av: *arcus vertebrae*, Ca: *vertebra caudalis*, cv: *corpus vertebrae*, eca: *extremitas caudalis*, ecr: *extremitas cranialis*, fiv: *foramen intervertebrale*, fv: *foramen vertebrale*, lav: *lamina arcus vertebrae*, paca: *processus articularis caudalis*, pacr: *processus articularis cranialis*, ps: *processus spinosus*, pt: *processus transversus*.

**Figure 16 animals-13-03124-f016:**
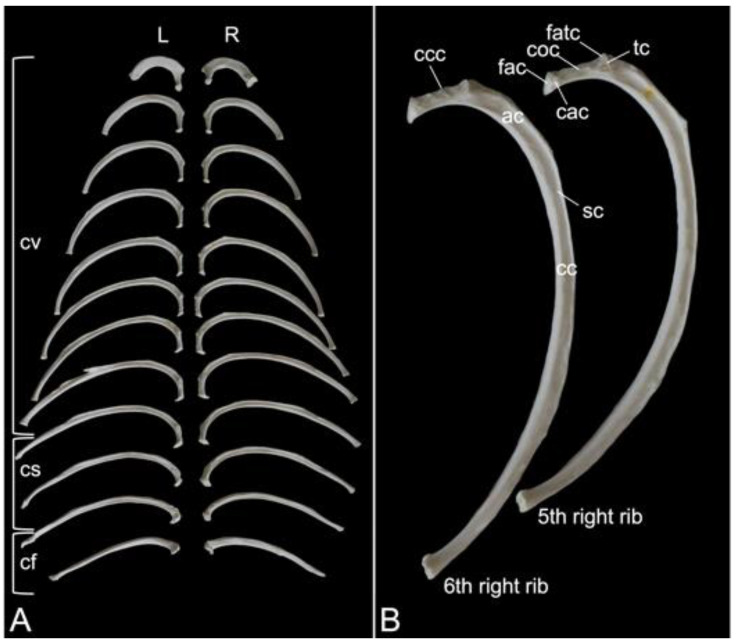
Ribs of the hamadryas baboon. (**A**) Both sets of 12 left (L) and right (R) ribs displayed with their medial aspects facing the viewer. (**B**) Medial aspects of the fifth and sixth right ribs. ac: *angulus costae*, cac: *caput costae*, cc: *corpus costae*, ccc: *crista colli costae*, cf: *costae fluctuantes*, coc: *collum costae*, cs: *costae spuriae*, cv: *costae verae*, fac: *facies articularis capitis costae*, fatc: *facies articularis tuberculi costae*, sc: *sulcus costae*, tc: *tuberculum costae*.

**Figure 17 animals-13-03124-f017:**
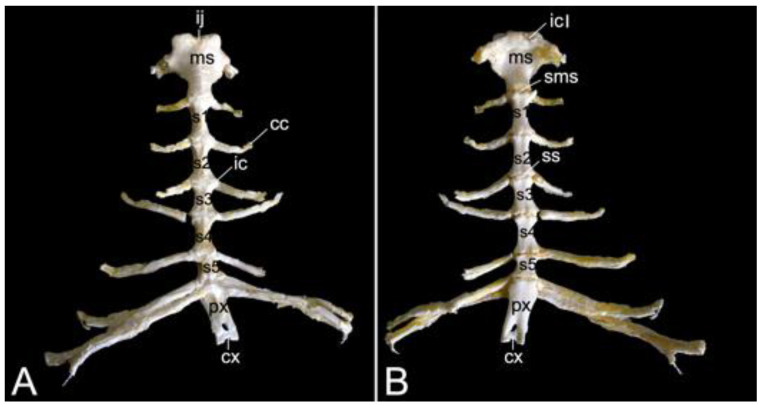
Sternum of the hamadryas baboon. (**A**) Ventral view, (**B**) Dorsal view. cc: *cartilago costalis*, cx: *cartilago xiphoidea*, ic: *incisura costalis*, icl: *incisura clavicularis*, ij: *incisura jugularis*, ms: *manubrium sterni*, px: *processus xiphoideus*, s: *sternebra*, sms: *synchondrosis manubriosternalis*, ss: *synchondrosis sternalis*.

**Figure 18 animals-13-03124-f018:**
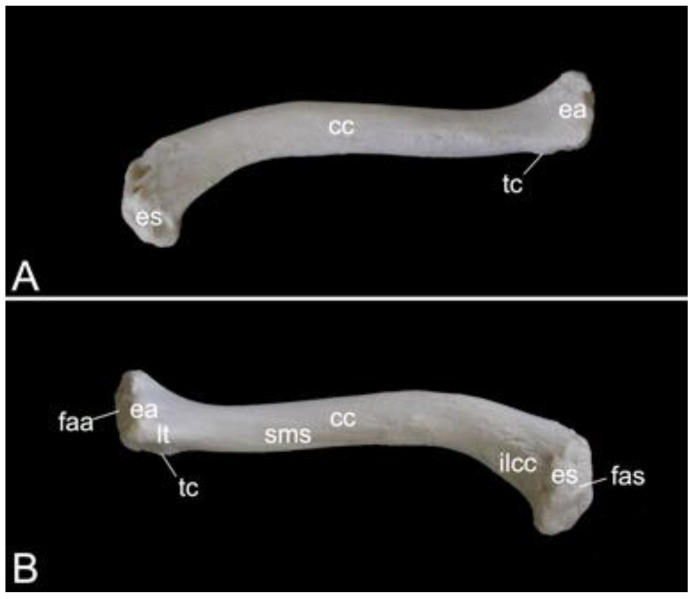
Left clavicle of the hamadryas baboon. (**A**) Cranial view, (**B**) Caudal view. cc: *corpus claviculae*, ea: *extremitas acromialis*, es: *extremitas sternalis*, faa: *facies articularis acromialis*, fas: *facies articularis sternalis*, ilcc: *impressio ligamenti costoclavicularis*, lt: *linea trapezoidea*, sms: *sulcus musculi subclavii*, tc: *tuberculum conoideum*.

**Figure 19 animals-13-03124-f019:**
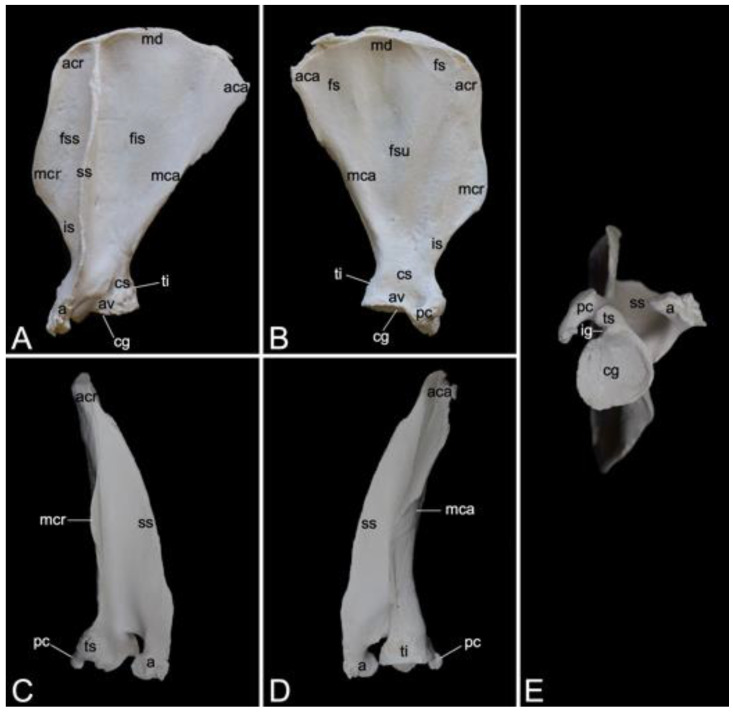
Left *scapula* of the hamadryas baboon. (**A**) Lateral view (*facies lateralis*), (**B**) Medial view (*facies costalis*), (**C**) Cranial view, (**D**) Caudal view, (**E**) Ventral view. a: *acromion*, aca: *angulus caudalis*, acr: *angulus cranialis*, av: *angulus ventralis*, cg: *cavitas glenoidalis*, cs: *collum scapulae*, fis: *fossa infraspinata*, fss: *fossa supraspinata*, fsu: *fossa subscapularis*, ig: *incisura glenoidalis*, is: *incisura scapulae*, mca: *margo caudalis*, mcr: *margo cranialis*, md: *margo dorsalis*, pc: *processus coracoideus*, ss: *spina scapulae*, ti: *tuberculum infraglenoidale*, ts: *tuberculum supraglenoidale*.

**Figure 20 animals-13-03124-f020:**
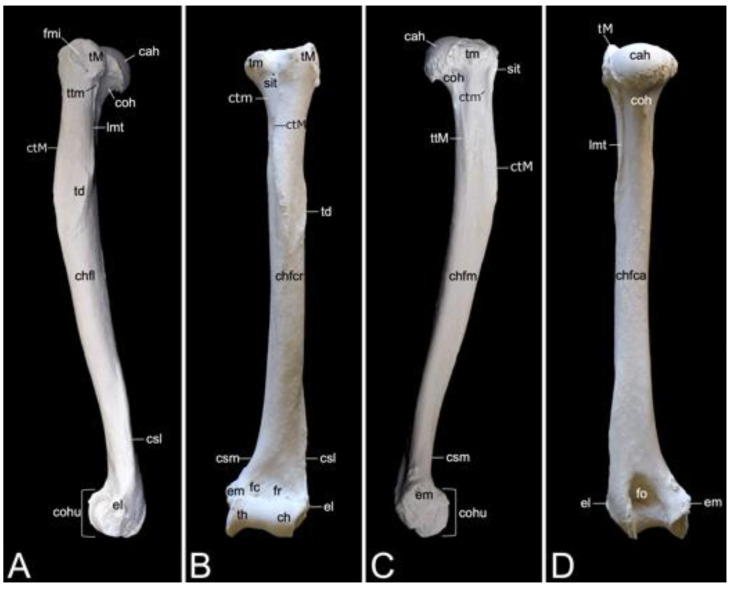
Left humerus of the hamadryas baboon. (**A**) Lateral view, (**B**) Cranial view, (**C**) Medial view, (**D**) Caudal view. cah: *caput humeri*, ch: *capitulum humeri*, chfca: *corpus humeri facies caudalis*, chfcr: *corpus humeri facies cranialis*, chfl: *corpus humeri facies lateralis*, chfm: *corpus humeri facies medialis*, coh: *collum humeri*, cohu: *condylus humeri*, csl: *crista supracondylaris lateralis*, csm: *crista supracondylaris medialis*, ctm: *crista tuberculi minoris*, ctM: *crista tuberculi majoris*, el: *epicondylus lateralis*, em: *epicondylus medialis*, fc: *fossa coronoidea*, fmi: *facies musculi infraspinati*, fo: *fossa olecrani*, fr: *fossa radialis*, llsit: *labium laterale sulci intertubercularis*, lmsit: *labium mediale sulci intertubercularis*, lmt: *linea musculi tricipitis*, sit: *sulcus intertubercularis*, td: *tuberositas deltoidea*, th: *trochlea humeri*, tM: *tuberculum majus*, tm: *tuberculum minus*, ttm: *tuberositas teres minor*.

**Figure 21 animals-13-03124-f021:**
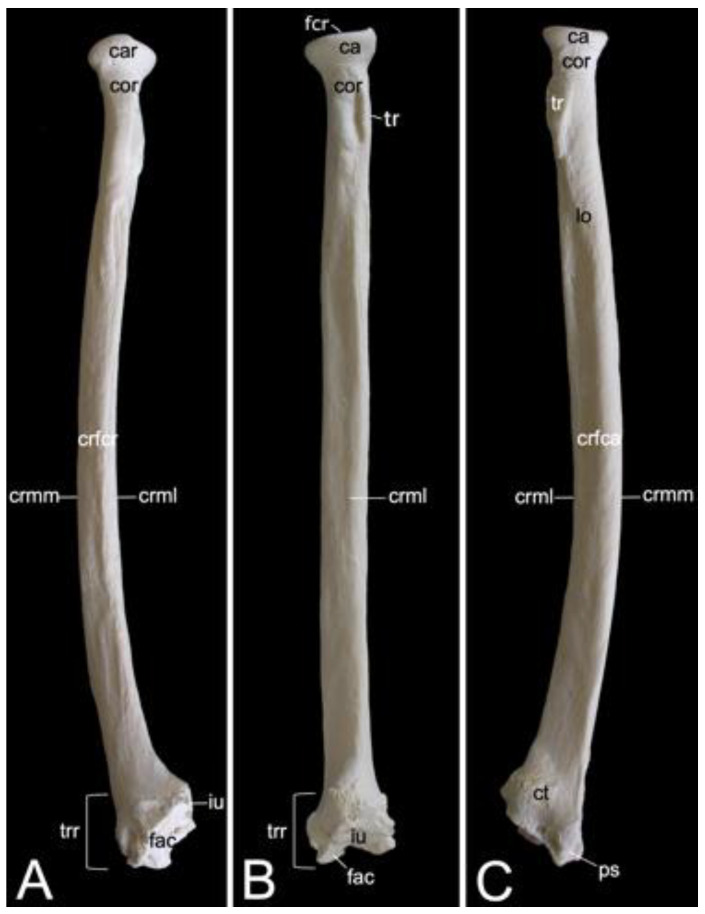
Left *radius* of the hamadryas baboon. (**A**) Cranial view, (**B**) Lateral view (**C**) Caudal view. ca: *circumferentia articularis*, car: *caput radii*, cor: *collum radii*, crfca: *corpus radii facies caudalis*, crfcr: *corpus radii facies cranialis*, crml: *corpus radii margo lateralis*, crmm: *corpus radii margo medialis*, ct: *crista transversa*, fac: *facies articularis carpea*, fcr: *fovea capitis radii*, iu: *incisura ulnaris*, lo: *linea obliqua*, ps: *processus styloideus*, tr: *tuberositas radii*, trr: *trochlea radii*.

**Figure 22 animals-13-03124-f022:**
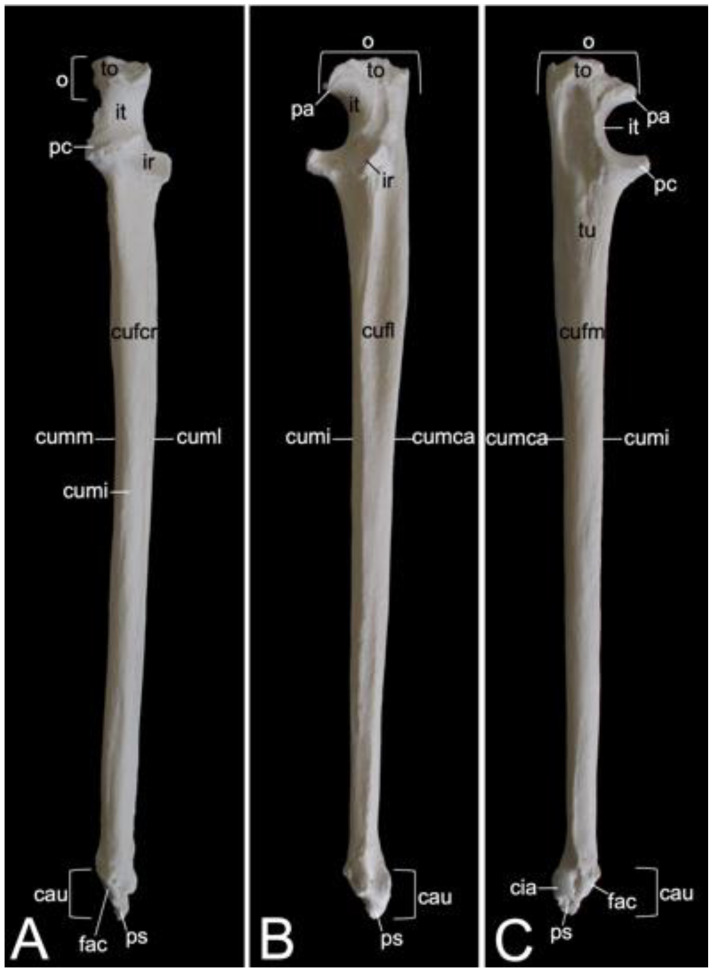
Left *ulna* of the hamadryas baboon. (**A**) Cranial view, (**B**) Lateral view (**C**) Medial view. cau: *caput ulnae*, cia: *circumferentia articularis*, cufm: *corpus ulnae facies medialis*, cufcr: *corpus ulnae facies cranialis*, cufl: *corpus ulnae facies lateralis*, cumca: *corpus ulnae margo caudalis*, cumi: *corpus ulnae margo interosseus*, cuml: *corpus ulnae margo lateralis*, cumm: *corpus ulnae margo medialis*, fac: *facies articularis carpea*, ir: *incisura radialis*, it: *incisura trochlearis*, o: *olecranon*, pa: *processus anconeus*, pc: *processus coronoideus*, ps: *processus styloideus*, to: *tuber olecrani*, tu: *tuberositas ulnae*.

**Figure 23 animals-13-03124-f023:**
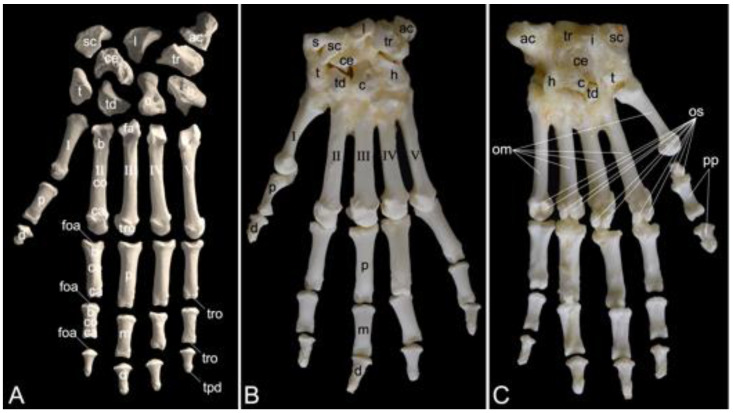
Skeleton of the left hand of the hamadryas baboon. (**A**) Dorsal view after complete maceration, (**B**) Dorsal view after partial maceration leaving the intercarpal and collateral ligaments in situ, (**C**) Palmar view after partial maceration leaving the intercarpal and collateral ligaments in situ. ac: *os pisiforme* (*os carpi accessorium*), b: *basis*, c: *os capitatum* (*os carpale tertium*), ca: *caput*, ce: *os carpi centrale*, co: *corpus*, d: *phalanx distalis*, fa: *facies articularis*, foa: *fovea articularis*, h: *os hamatum* (*os carpale quartum*), I: *os metacarpale primum*, II: *os metacarpale secundum*, III: *os metacarpale tertium*, IV: *os metacarpale quartum*, l: *os lunatum* (*os carpi intermedium*), m: *phalanx media*, om: *ossa metacarpalia*, os: *ossa sesamoidea palmaria*, p: *phalanx proximalis*, pp: *phalanges pollicis*, s: *os sesamoideum m. abductoris digiti primi / pollicis longi*, sc: *os scaphoideum* (*os carpi radiale*), t: *os trapezium* (*os carpale primum*), td: *os trapezoideum* (*os carpale secundum*), tpd: *tuberositas phalangis distalis*, tr: *os triquetrum* (*os carpi ulnare*) tro: *trochlea*, V: *os metacarpale quintum*.

**Figure 24 animals-13-03124-f024:**
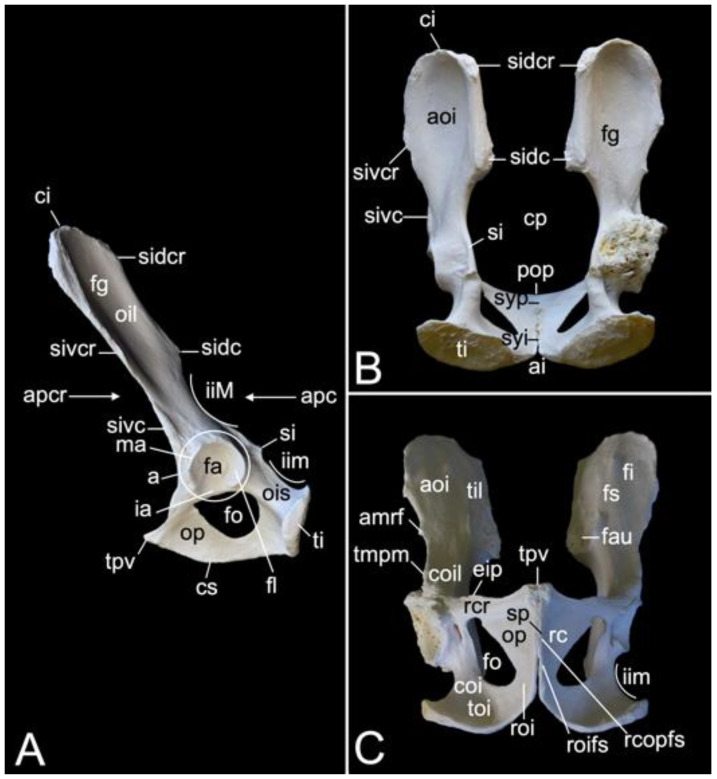
Hip bone and hip (the sacrum is not shown) of the hamadryas baboon. (**A**) Lateral view of the left hip bone, (**B**) Caudal view of the hip, (**C**) Ventral view of the hip. Notice the arthritic right *acetabulum*. a: *acetabulum*, ai: *arcus ischiadicus*, amrf: *area musculi recti femoris*, aoi: *ala ossis ilii*, apc: *apertura pelvis caudalis*, apcr: *apertura pelvis cranialis*, ci: *crista iliaca*, coi: *corpus ossis ischii*, coil: *corpus ossis ilii*, cp: *cavum pelvis*, cs: *crista symphysialis*, eip: *eminentia iliopubica*, fa: *fossa acetabuli*, fau: *facies auricularis*, fg: *facies glutea*, fi: *fossa iliaca*, fl: *facies lunata*, fo: *foramen obturatum*, fs: *facies sacropelvina*, ia: *incisura acetabuli*, iiM: *incisura ischiadica major*, iim: *incisura ischiadica minor*, ma: *margo acetabuli*, oil: *os ilium*, ois: *os ischium*, op: *os pubis*, pop: *pecten ossis pubis*, rc: *ramus caudalis ossis pubis*, rcopfs: *ramus caudalis ossis pubis facies symphysialis*, rcr: *ramus cranialis ossis pubis*, roi: *ramus ossis ischii*, roifs: *ramus ossis ischii facies symphysialis*, si: *spina ischiadica*, sidc: *spina iliaca dorsalis caudalis*, sidcr: *spina iliaca dorsalis cranialis*, sivc: *spina iliaca ventralis caudalis*, sivcr: *spina iliaca ventralis cranialis*, sp: *symphysis pelvina*, syi: *symphysis ischiadica*, syp: *symphysis pubica*, ti: *tuber ischiadicum*, til: *tuberositas iliaca*, tmpm: *tuberculum musculi psoas minoris*, toi: *tabula ossis ischii*, tpv: *tuberculum pubicum ventrale*.

**Figure 25 animals-13-03124-f025:**
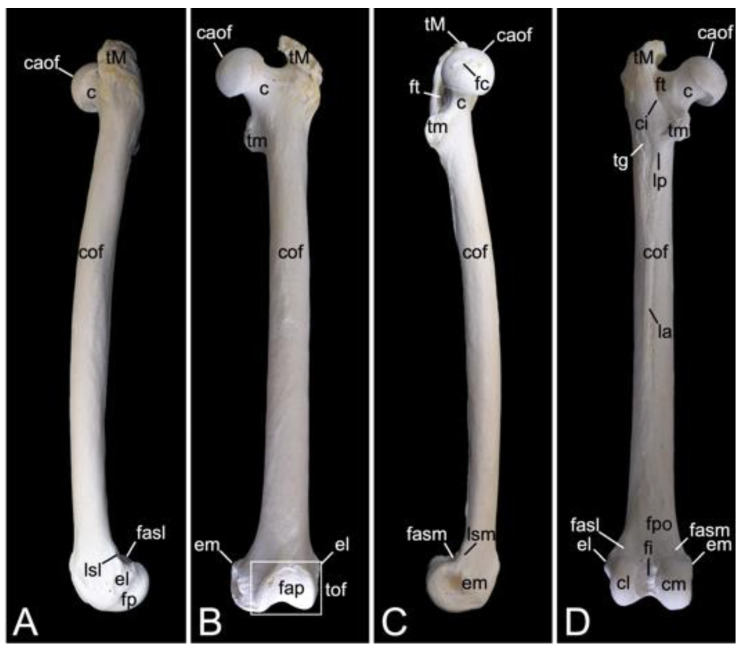
Left femur of the hamadryas baboon. (**A**) Lateral view, (**B**) Cranial view, (**C**) Medial view, (**D**) Caudal view. c: *collum ossis femoris*, caof: *caput ossis femoris*, ci: *crista intertrochanterica*, cl: *condylus lateralis*, cm: *condylus medialis*, cof: *corpus ossis femoris*, el: *epicondylus lateralis*, em: *epicondylus medialis*, fap: *facies articularis patellaris*, fasl: *facies articularis sesamoidea lateralis*, fasm: *facies articularis sesamoidea medialis*, fc: *fovea capitis*, fi: *fossa intercondylaris*, fp: *fossa musculi poplitei*, fpo: *facies poplitea*, ft: *fossa trochanterica*, la: *linea aspera*, lp: *linea pectinea*, lsl: *linea supracondylaris lateralis*, lsm: *linea supracondylaris medialis*, tg: *tuberositas glutea*, tM: *trochanter major*, tm: *trochanter minor*, tof: *trochlea ossis femoris*.

**Figure 26 animals-13-03124-f026:**
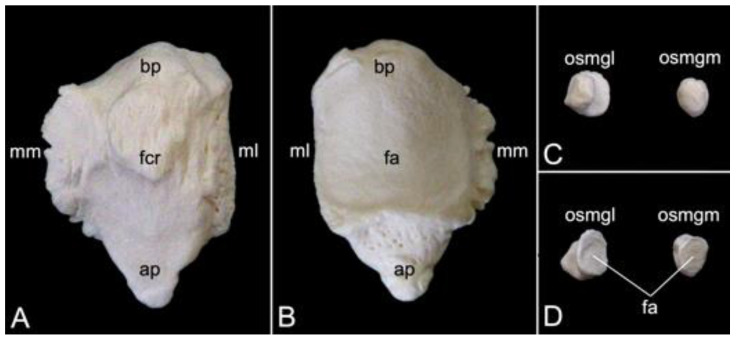
Sesamoid bones associated with the left femur of the hamadryas baboon. (**A**) Cranial view of the *patella*, (**B**) Caudal view of the *patella*, (**C**) Caudal view of the sesamoid bones of the gastrocnemius muscle, (**D**) Cranial view of the sesamoid bones of the gastrocnemius muscle. ap: *apex patellae*, bp: *basis patellae*, fa: *facies articularis*, fcr: *facies cranialis*, ml: *margo lateralis*, mm: *margo medialis*, osmgl: *os sesamoideum musculi gastrocnemii laterale*, osmgm: *os sesamoideum musculi gastrocnemii mediale*.

**Figure 27 animals-13-03124-f027:**
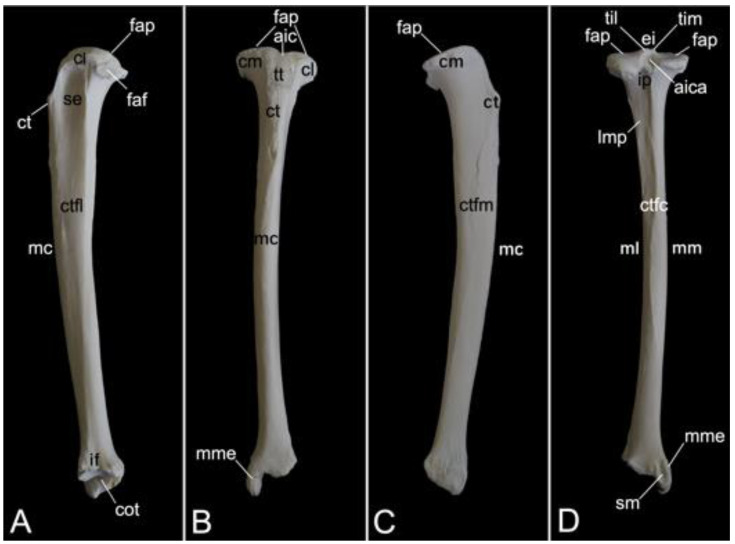
Tibia of the hamadryas baboon. (**A**) Lateral view, (**B**) Cranial view, (**C**) Medial view, (**D**) Caudal view. aic: *area intercondylaris cranialis*, aica: *area intercondylaris caudalis*, cl: *condylus lateralis*, cm: *condylus medialis*, cot: *cochlea tibiae*, ct: *crista tibiae*, ctfc: *corpus tibiae facies caudalis*, ctfl: *corpus tibiae facies lateralis*, ctfm: *corpus tibiae facies medialis*, ei: *eminentia intercondylaris*, faf: *facies articularis fibulae*, fap: *facies articularis proximalis*, if: *incisura fibularis*, ip: *incisura poplitea*, lmp: *linea musculi poplitei*, mc: *margo cranialis*, ml: *margo lateralis*, mm: *margo medialis*, mme: *malleolus medialis*, se: *sulcus extensorius*, sm: *sulcus malleolaris*, til: *tuberculum intercondylare laterale*, tim: *tuberculum intercondylare medialie*, tt: *tuberositas tibiae*.

**Figure 28 animals-13-03124-f028:**
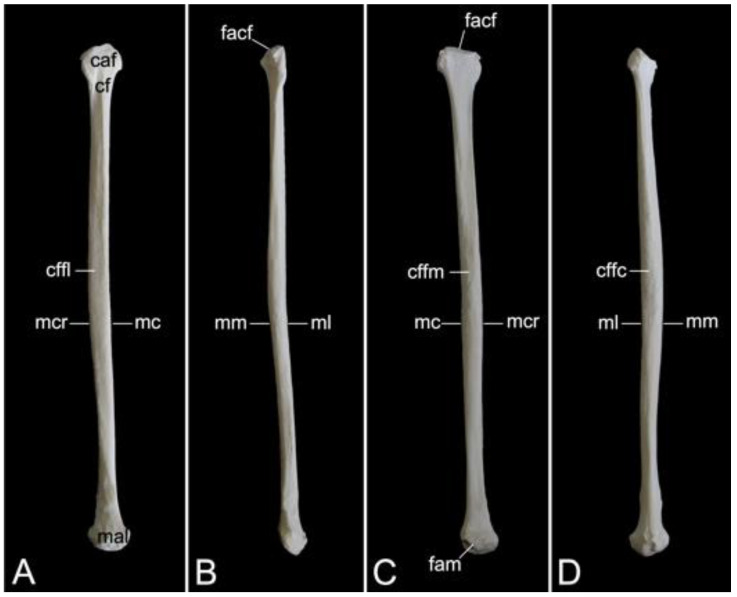
Fibula of the hamadryas baboon. (**A**) Lateral view, (**B**) Cranial view, (**C**) Medial view, (**D**) Caudal view. caf: *caput fibulae*, cf: *collum fibulae*, cffc: *corpus fibulae facies caudalis*, cffl: *corpus fibulae facies lateralis*, cffm: *corpus fibulae facies medialis*, facf: *facies fibularis capitis fibulae*, fam: *facies articularis malleoli*, mal: *malleolus lateralis*, mc: *margo caudalis*, mcr: *margo cranialis*, ml: *margo lateralis*, mm: *margo medialis*.

**Figure 29 animals-13-03124-f029:**
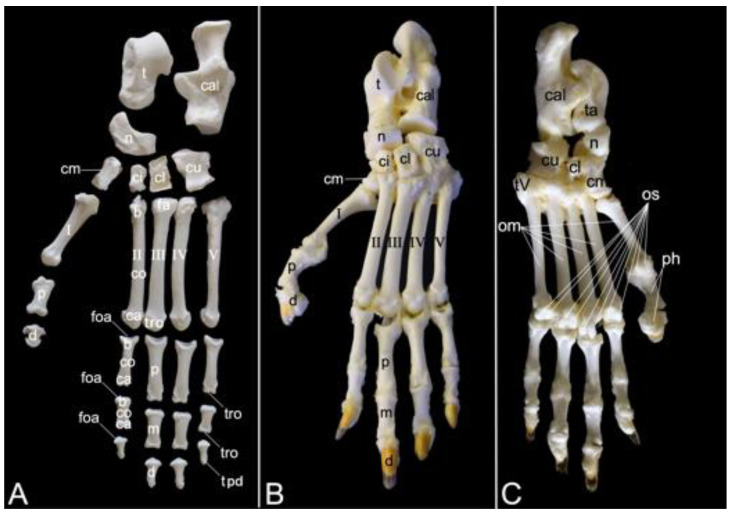
Left foot of the hamadryas baboon. (**A**) Dorsal view after complete maceration, (**B**) Dorsal view after partial maceration leaving the intertarsal and collateral ligaments in situ, (**C**) Plantar view after partial maceration leaving the intertarsal and collateral ligaments in situ. b: *basis*, ca: *caput*, cal: *calcaneus*, ci: *os cuneiforme intermedium* (*os tarsale secundum*), cl: *os cuneiforme laterale* (*os tarsale tertium*), cm: *os cuneiforme mediale* (*os tarsale primum*), co: *corpus*, cu: *os cuboideum* (*os tarsale quartum*), d: *phalanx distalis*, fa: *facies articularis*, foa: *fovea articularis*, I: *os metatarsale primum*, II: *os metatarsale secundum*, III: *os metatarsale tertium*, IV: *os metatarsale quartum*, m: *phalanx media*, n: *os naviculare* (*os tarsi centrale*), om: *ossa metatarsalia*, os: *ossa sesamoidea plantaria*, p: *phalanx proximalis*, ph: *phalanges hallucis*, t: *talus*, tpd: *tuberositas phalangis distalis*, tro: *trochlea*, tV: *tuberositas ossis metatarsalis V*, V: *os metatarsale quintum*.

**Figure 30 animals-13-03124-f030:**
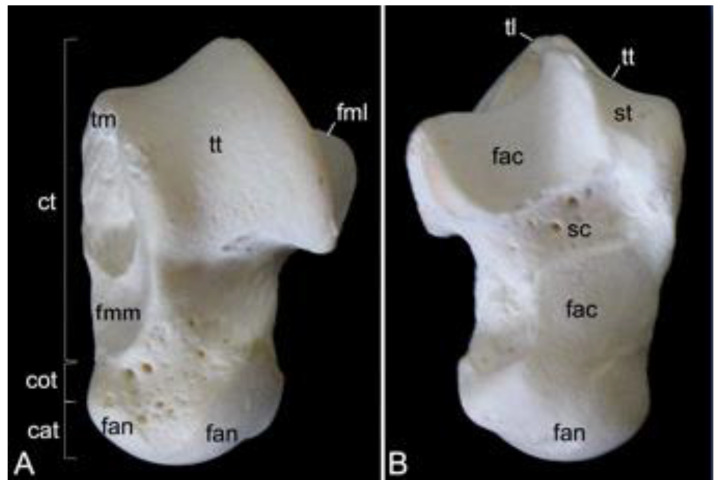
Left talus of the hamadryas baboon. (**A**) Dorsal view, (**B**) Plantar view. cat: *caput tali*, cot: *collum tali*, ct: *corpus tali*, fac: *facies articulares calcaneae*, fan: *facies articularis navicularis*, fml: *facies articularis malleolaris lateralis*, fmm: *facies articularis malleolaris medialis*, sc: *sulcus calcanei*, st: *sulcus tendinis*, tl: *tuberculum laterale*, tm: *tuberculum mediale*, tt: *trochlea tali*.

**Figure 31 animals-13-03124-f031:**
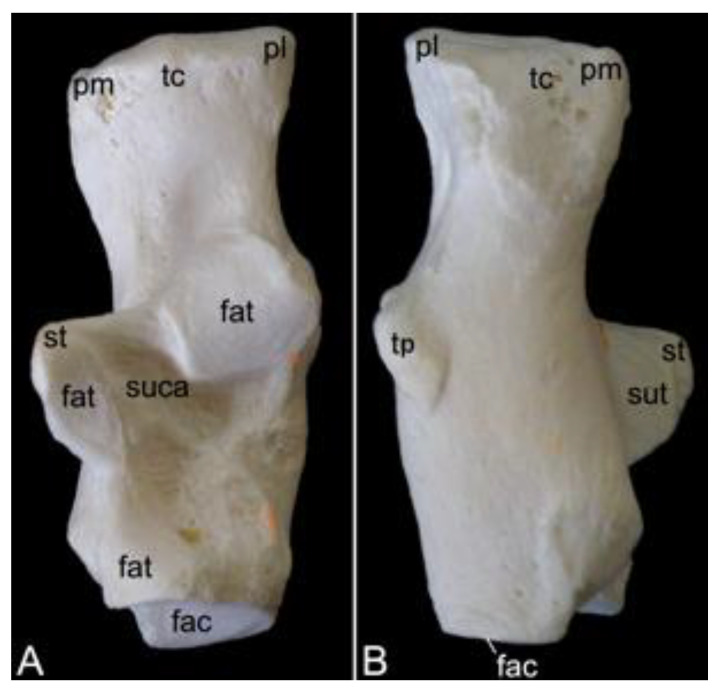
Left calcaneus of the hamadryas baboon. (**A**) Dorsal view, (**B**) Plantar view. fac: *facies articularis cuboidea*, fat: *facies articulares talares*, pl: *processus lateralis*, pm: *processus medialis*, st: *sustentaclum tali*, suca: *sulcus calcanei*, sut: *sulcus tendinis*, tc: *tuber calcanei*, tp: *tuber plantare*.

## Data Availability

Not applicable.

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
