# Peer review of "Osteology of the Hamadryas Baboon (Papio hamadryas)"

_animals, 2023, doi:10.3390/ani13193124_

Round 1
Reviewer 1 Report
The authors of the manuscript "Osteology of the hamadryas baboon (Papio hamadryas)" undertook a synthetic description of the baboon skeleton. This is undoubtedly an important book that complements knowledge in the field of comparative osteology. The importance of primates has recently been increasing, among other things, due to the development of translational medicine. The available literature is sparse and the anatomical nomenclature is disordered. Therefore, I appreciate the efforts of the authors who prepared a detailed description of the baboon skeleton, with particular emphasis on the skull. The text is clear, understandable and linguistically correct. The descriptions are supplemented with high-quality illustrations.
In the morphological description of primates, pitfalls can be expected, especially related to directional terminology. The authors try to reconcile Nomina Anatomica Veterinaria with human nominations, but sometimes it turns out to be difficult. For example, in the head skeleton of animals there are presphenoid and basosphenoidal bones. The authors use similar names in the text, but they also mention "sphenoid" as in a human in which both segments have fused. It is worth standardizing this.
Other specific comments:
Avoid "upper arm", "upper leg", "lower leg". Apply distal limb, stylopodium, zeugopodium, antebrachium etc.
Avoid "front limb", "hind limb", especially in relation to primates.
Line 212 - bony pelvis
Line 237 - facies is part of the body, while ossa faciei is part of the skeleton of the head
Line 585 - brain
Lines 587-589 - "internal aspect of the base of the skull" or "internal cranial base " - be consistent
Latin names of non-osteological structures like bulbus olfactorius, tractus olfactorius, nervus olfactorius, vermis cerebellaris are redundant
Line 700 - for. symphysialis
Avoid double brackets and instead write Latin names in italics after a comma, e.g. Line 1005 Sacrum, os sacrum (vertebrae sacrales)
Line 1061 - Thorax is a topographic name (body part) so it is not the same as skeleton thoracis
Line 1066 - These are parts of the bony rib (osa coste). Cartilogo costalis test also part of the rib (costa)
Line 1140 - Collar bone (clavicle), clavicula
Line 1160 - Shoulder blade, scapula
Line 1277 Skeleton antebtrachii:
Line 1366 - Skeleton manus (hand, manus is part of the body)
Line 1441 - Hip bone , os coxae
Line 1532 - Fermur, os femoris
Line 1711 - Skeleton pedis (foot is a part of the body)
Latin names should be italicized
Reviewer 2 Report
The paper concerns osteological studies of baboon hamadryas. Ostology these days is a little appreciated branch of anatomy hence I congratulate you for taking up this subject. In the introduction, the authors rightly point out that baboons are sometimes model animals for human medicine research.
- It should be added in the introduction that relationships between humans and baboons have varied throughout history and osteological studies can be extremely valuable for research on, for example, baboon mummies from Ancient Egypt
The authors rightly point out the disorder regarding anatomical nomenclature prevailing in publications on apes including baboons.
It is unfortunate that the study was carried out only on males and I believe that in the future it would be appropriate to compare the skeleton of females and males on the basis of certain parameters (e.g. von den Driesch 1976).
The description of the skull is extremly accurate and very well illustrated. The authors have also provided information that may also be clinically relevant (e.g. Bold J., Szemet M. et al. 2023).
I believe that when mentioning cranial nerves it is not necessary to state which cranial nerve it is because if you write e.g. n. glossopharyngeus it is known that it is the IX nerve - this lengthens the text unnecessarily.
It would make sense to write Latin names in italics- I am very pleased that authors use Latin names.
3.5.1. The number of circles should be given using numerals or Arabic letters, but this should be standardised for ease of reference- wskazane byłoby badanie na większej ilości zwierząt.
In the case of the number of vertebrae in the thoracic and lumbar segments, it should be indicated that this applies to these specimens, as these segments are characterised by variation.
The work is unfailingly solid and interesting. The excellent and well-described illustrations deserve to be highlighted.
Given the widespread use of radiography and computed tomography in veterinary medicine, it would be worthwhile to use these methods to image the baboon skeleton as well.
The literature is adequate but could be supplemented with items indicating the great importance of knowing the baboon skeleton in a broader context than just translational medicine.
